# Comparison of Generalized Non-Data-Driven Lake and Reservoir Routing Models for Global-Scale Hydrologic Forecasting of Reservoir Outflow at Diurnal Time Steps

Joseph L. Gutenson[1*], Ahmad A. Tavakoly[1,2], Mark D. Wahl[1], Michael L. Follum[1]

[1]Research Civil Engineer (Hydraulics), US Army Engineer Research and Development Center, Coastal and Hydraulics Laboratory, 3909 Halls Ferry Rd., Vicksburg, MS 39056

[2]University of Maryland, Earth System Science Interdisciplinary Center, College Park, Maryland, USA

[*]Corresponding Author: jlgutenson@gmail.com

Abstract: Large-scale hydrologic forecasts should account for attenuation through lakes and reservoirs when flow regulation is present. Globally generalized methods for approximating outflow are required but must contend with operational complexity and a dearth of information on dam characteristics at global spatial scales. There is currently no consensus on the best approach for approximating reservoir release rates in large spatial scale hydrologic forecasting, particularly at diurnal time steps. This research compares two parsimonious reservoir routing methods at daily steps; Döll et al. (2003) and Hanasaki et al. (2006). These reservoir routing methods have been previously implemented in large-scale hydrologic modeling applications and have been typically evaluated seasonally. These routing methods are compared across 60 reservoirs operated by the U.S. Army Corps of Engineers. The authors vary empirical coefficients for both reservoir routing methods as part of a sensitivity analysis. The method proposed by Döll et al. (2003) outperformed that presented by Hanasaki et al. (2006) at a daily time step and improved model skill over most run-of-the-river conditions. The temporal resolution of the model influences models performances. The optimal model coefficients varied across the reservoirs in this study and model performance fluctuates between wet years and dry years, and for different configurations such as dams in series. Overall, the method proposed by Döll et al. (2003) could enhance large scale hydrologic forecasting, but can be subject to instability under certain conditions.

# 1. Introduction

## 1.1. Importance of Dams in Hydrologic Simulations

Improvements in numerical weather prediction, the increasing abundance of computational power, and greater precision of remotely sensed observations make global hydrologic forecasting and flood warning systems increasingly feasible (Alfieri et al., 2013; Wu et al., 2014; Emerton et al., 2016; Salas et al., 2017). Lack of information concerning anthropogenic influences on runoff is a major deficiency of large-scale flood forecasting systems (Emerton et al., 2016). Reservoir operations tend to distort natural flow patterns, effectively redistributing surface water spatially and temporally (Zhou et al., 2016). Impoundments significantly influence the downstream flow regime at small and large spatial scales (Batalla et al., 2004; Magilligan and Nislow, 2005). Over half of the world's large river systems are now substantially altered by dams (Nilsson et al., 2005) resulting in a seven-fold increase in water storage within the global river system (Vörösmarty et al. 1997). Furthermore, the cumulative alterations from global reservoir impoundments are so significant that it has been suggested that they could buffer global sea-level rise (Chao et al., 2008).

Dams primarily impact the hydrologic cycle by changing the magnitude and timing of the discharges downstream (Haddeland et al., 2006; Döll et al., 2009; Biemans et al., 2011; Wu et al., 2014; Zajac et al., 2017), often with the specific intent to mitigate hydrologic extremes (i.e., floods and droughts) (Zajac et al., 2017). Dams reduce peak discharges by roughly a third on average while dampening the daily variation by a similar amount (Graf, 2006). In hydrologic forecasting, accuracy of the timing and magnitude of hydrologic extremes is fundamentally important to the usefulness of the forecasts.

Therefore, the significant impacts from dams make inclusion of reservoir operations, or
reservoir routing, critical in large scale hydrologic flood forecasting.

Integrating dam operations within large-scale river routing and flood forecasting

improves model performance downstream of reservoir locations (Snow et al., 2016;
Tavakoly et al., 2017; Salas et al., 2017; Zajac et al., 2017). This is often not feasible at
large-scales since there may be multiple entities responsible for regulating flow,
particularly with respect to transboundary waters. Among other things, operational
knowledge, site-specific rule curves, reservoir uses, and local decision-making practices at
each individual project dictate dam releases. Thus, dam operations are typically non-linear,
complex processes, driven by anthropogenic and environmental influences. This makes
generalizing reservoir operations difficult, particularly in the context of predicting dam-
induced hydrologic responses at diurnal or sub-diurnal time step. Heuristically accounting
for dams within existing routing schemes should improve flood forecast results when
scheduled releases are not readily known.

Reservoir routing methodologies are generally divided into two basic categories:

data-driven and non-data-driven. Machine-learning, artificial intelligence (Coerver et al.,
2017; Macian-Sorribes and Pulido-Velazquez, 2017; Ehsani et al., 2016; Mohan and
Ramsundram, 2016; Ticlavilca and McKee, 2011; Chaves and Chang, 2008; Khalil et al.,
2005), and remote sensing (Bonnema et al., 2016; Yoon and Beighley, 2015) are examples
of data-driven approaches. Such data-driven methodologies can be effectively applied to
dynamic non-linear systems, particularly when the governing influence on the system does
not follow any particular deterministic model. These types of approaches require training
data or specific knowledge of a particular reservoir to effectively parameterize and apply
them. This is often an insurmountable limitation for data-driven approaches. For that
reason, the focus of this paper is on non-data-driven reservoir routing methodologies as an
incremental improvement over schemes that effectively neglect dams when information is
scarce.
1.2.    Non-Data-Driven Reservoir Storage and Outflow Simulation

Non-data-driven approaches to reservoir routing rely on conceptualizing reservoir

responses without explicitly observing the actual reservoir operations. The optimal method
for a given application depends on a balance between complexity and available information
(De Vos, 2015). Therefore, this manuscript focuses on selecting for parsimony.

Existing non-data-driven reservoir models range from simple approaches to

sophisticated methods. Solander et al. (2016) showed that temperature-based schema best
fits the modeling of discharge, $Q_{out,t}$. The Solander et al. (2016) rule is driven by
temperature shifts at each model time step above and below the mean temperature. The
Solander et al. (2016) method indicates that temperature is the main proxy governing
reservoir release, due to the assumption that seasonality drives agricultural production and
reservoir operation. However, the Solander et al. (2016) study focuses on long-term
climatic forecasting. Diurnal temperature variations will not likely describe day-to-day
reservoir operations. Zhao et al., (2016) developed a reservoir routing scheme based on
reservoir stage and storage rules. However, real-time insights related to current reservoir
stages throughout a region can involve considerable remotely sensed information. The
stage information must then be related somehow to storage volume making this a much
more data-driven process. Burek et al. (2013) also developed a non-data-driven approach
to reservoir routing which was implemented by Zajac et al. (2017). This approach is built
into the LISFLOOD model. The Burek et al. (2013) model requires a number of
assumptions about storage capacity limits and naturalized streamflow thresholds. For
example, the minimum, normal, and maximum storage are assumed to be 0.1, 0.3, and
0.97, respectively. To maintain the objective of investigating parsimonious models, the
approach by Burek et al. (2013) was not included in this evaluation.

Döll et al. (2003), Wada et al. (2014), and Wisser et al. (2010) presented non-data-

driven methods to simulate reservoirs operation that can be considered as simple
approaches. The Wisser et al. (2010) method follows a simple, rule-based approach to
define the reservoir outflow at each time step ($Q_{out,t}$). The rule that Wisser et al. (2010)
enacts is that when the inflow at each model time step moves above or below the long-term
average inflow, the behavior of the reservoir release changes. De Vos (2015) suggested
that this model is too simple to effectively model reservoir outflow. In a similar vein, Wada
et al. (2014) introduced a daily estimate of reservoir outflow that is simply the product of
the proportion of available reservoir storage and daily inflow, which can be too simplistic
to estimate reservoir outflow since no coefficient is introduced into the simulation to
account for reservoir heterogeneity.

Döll et al. (2003) derived reservoir routing scheme that can be applied to man-made

reservoirs and natural water bodies. The Döll et al. (2003) methodology found genesis in
the reservoir outflow model proposed by Meigh et al. (1999). Meigh et al. (1999) proposed
a simple reservoir release methodology, which intended to mimic outflow at reservoirs
from a theoretical rectangular weir. A more substantive version of the Meigh et al. (1999)
method is formulated by Döll et al. (2003). Despite its simplicity, the Döll et al. (2003)
method demonstrated good performance compared to several other routing methods (De
Vos, 2015).  The form of the Döll et al. (2003) equation is similar to that proposed by Wada
et al. (2014).  However, the Döll et al. (2003) methodology incorporates a coefficient that
can incorporate a portion of reservoir heterogeneity.

Compared to the aforementioned methods, Hanasaki et al. (2006) derived a demand

driven approach to reservoir routing, which can be considered a complicated non-data-
driven reservoir routing model. They distinguished between irrigation and non-irrigation
reservoirs and offered two distinct algorithms for each. Water demands for irrigation,
domestic, and industrial uses are considered in the irrigation reservoirs, whereas the
releases from non-irrigation reservoirs are simply a proportion of inflow.

De Vos (2015) also proposed a within-year/over-year reservoir routing method

comprised of two systems of equations, which was considered a non-data-driven approach.
Within-year reservoir operations are driven by yearly fill and release cycles and typically
have a small storage capacity relative to their total annual demand. Thus, water
accumulates during wet periods and decreases during dry periods. Over-year reservoir
operation, on the other hand, is based on long-term, multi-year drawdowns. Over-year
reservoirs have storage which is sufficiently large, relative to inflow, so that yearly cycles
of water storage and release are not necessary (Adeloye and Montaseri, 2000; Vogel et al.,
1999). De Vos (2015) compared his methodology to the Hanasaki et al (2006), Döll et al.
(2003), and Neitsch et al. (2011).  The De Vos (2015) over-year simulation assumes
knowledge of the mean and standard deviation of reservoir storage and is still too data-
driven for the purposes of this study.  Table 1 summarizes each of the inputs required by
each non-data-driven approach described above.

Table 1. Input requirements for the various reservoir routing methods.

| | Burek et al. (2013) | Zhao et al. (2016) | De Vos (2015) | Solander et al. (2016) | Döll et al. (2003) | Hanasaki et al. (2006) Non-irrigation Method | Wisser et al. (2010) | Wada et al. (2014) |
|---|---|---|---|---|---|---|---|---|
| Reservoir Inflow at time step | X | X | | X | X | X | X | X |
| Empirical Coefficients | | X | | X | X | X | X | |
| Minimum Storage/Inactive Storage Limit | X | X | X | | X | X | | X |
| Maximum Storage/Flood Storage Limit | X | X | X | | X | X | | X |
| Average Storage | | | X | | | | | |
| Standard Deviation of Storage | | | X | | | | | |
| Water Stored at model time step | X | X | | X | X | | | |
| Average Inflow | X | | X | | | X | X | |
| Flood Inflow | | X | | | | | | |
| Air Temperature | | | | X | | | | |
| Conservation Storage Limit | | X | | | | | | |
| Normal Storage Limit | X | | | | | | | |
| Normal Outflow | X | | | | | | | |
| Non-Damaging Outflow | X | | | | | | | |
| Precipitation on the Reservoir | X | | | | | | | |
| Evaporation From the Reservoir | X | | | | | | | |
| Fill Fraction | X | | | | | | | |
| Average Total Winter Inflow | | | | X | | | | |
| Pool Elev. at model time step | | X | | | | | | |
| Pool Elev. at top of inactive storage | | X | | | | | | |
| Pool Elev. at the top of conservation storage | | X | | | | | | |
| Pool Elev. at the top of flood storage | | X | | | | | | |
| Flood Seasonality | | | X | | | | | |
| Standardized Precipitation Evapotranspiration Index | | | X | | | | | |


The Döll et al. (2003) and Hanasaki et al. (2006) require minimal input data to
implement: reservoir inflow, average inflow, and storage volume characteristics. Each of
these variables are available in existing datasets, such as the Global Reservoir and Dam
(GRanD) database (Lehner et al., 2011) or can be generated using climate reanalysis data
(Snow et al., 2016). Other non-data-driven methods require data inputs that are not globally
available or produced within the hydrologic simulation (De Vos, 2015; Zhao et al., 2016;
Burek et al., 2013; Zajac et al., 2017). For example, the Global Flood Awareness System
(GloFAS) is the only existing, operational flood forecasting system that accounts for
reservoirs at continental to global spatial extents. However, the reservoir routing
component of GloFAS requires operational assumptions be made because of a lack of
global reservoir operational records (Zajac et al., 2017). Döll et al. (2003) (hereafter
referred to as D03) and Hanasaki et al. (2006) (hereafter referred to as H06) do not require
that these assumptions be made because of the minimal inputs which they require. Thus,
D03 and H06 meet the requirements of being parsimonious with respect to available
reservoir information.

The Döll et al. (2003) and Hanasaki et al. (2006) methods also provide enough

complexity to account for a portion of the model complexity inherent in reservoir
operations. De Vos (2015) does not employ the reservoir routing approach of Wisser et al.
(2010) because De Vos (2015) and neither does this research, as it does not account for the
status of the reservoir at each simulation time step. The approach taken by Wada et al.
(2014) is similar to D03 but represents reservoirs with similar inflow and storage
characteristics homogeneously.

Furthermore, D03 and H06 methods have been implemented in large-scale

hydrologic models. D03 was used in the WaterGAP model and the application of H06 was
implemented in the TRIP model by the same authors. The main difference in this
evaluation and previous evaluations (i.e., Hanasaki et al., 2006; Masaki et al., 2017) of
these reservoir routing schemes is that this research evaluates model performance at a
diurnal time step.

The aim of this study is to assess non-data-driven reservoir routing methods that

are parsimonious and align with available information for use in hydrologic forecasting
schemes applicable across the global domain at diurnal time steps. Considering these
research aims, the non-data driven reservoir routing methods developed by Döll et al.
(2003) and Hanasaki et al. (2006) were considered.

The following research questions are addressed with respect to the D03 and H06

approaches: (1) How well do the selected reservoir routing models improve outflow
estimates relative to simulation of naturalized flow (i.e. neglecting dams altogether)? (2)
How do reservoir routing coefficients affect model performance? (3) How does the time
step affect model performance and stability? This is a critical point for the current regional-
to continental-scale forecasting schemes that operate at daily or sub-daily time steps. (4)
How sensitive are the reservoir routing schemes to various real-world dam operations and
climate variability?

To achieve the research objectives of the study, reservoir data including daily

inflow and outflow from 2006-2012, for 60 U.S. Army Corps of Engineers (USACE)
reservoirs were used to evaluate the reservoir routing schemes. The data were obtained
from nine USACE districts: Pittsburg, Nashville, St. Paul, Rock Island, Omaha, Tulsa,
Sacramento, Los Angeles, and Vicksburg. The selected dams are representative of a wide
range of reservoir sizes, flow regimes, and climatologic settings but are predominately
managed for flood control. The results of this analysis will benefit readers in determining
if the reservoir routing models implemented within existing, large-scale hydrologic
forecasts adequately represent reservoir effects.

2.  Methodology

2.1.    Simulation Specifications

The storage ratio (Vogel et al., 1999) or Impoundment Ratio is an important metric

in previous works examining generalizing reservoir operation (De Vos, 2015; Hanasaki et
al., 2006). The impoundment ratio is described as follows:
$IR = \frac{(S_{max} - S_{min})}{Q_{in} * 86400 * 365}$                              (1)
where $S_{max}$ and $S_{min}$ are the maximum and minimum volumes of the reservoir's active
storage [m3], and $Q_{in}$ is the mean annual inflow to the reservoir [m3s-1].

A higher impoundment ratio indicates that the capacity of the reservoir is large

relative to mean inflows, while the opposite is true of low IR values.  De Vos (2015)
considered IR values greater than unity "large" reservoirs, as they are capable of storing
the average yearly volume of water flowing into them. To utilize H06, the release
coefficient ($k_r$) needs to be determined.
$k_r = \frac{S_{begin}}{\alpha S_{max}}$                              (2)
where $S_{begin}$ is the storage [m$^3$] at the beginning of each year and $\alpha$ is a dimensionless
coefficient, which was set to 0.85 in the Hanasaki et al. (2006) study. In the current study,
the $\alpha$ parameter was varied from 0.45-0.95 by increments of 0.10 and solve $k_r$ for each $\alpha$
value.
Outflow is the quantity of most interest for hydrologic flood forecasting because
these forecasts generally occur over a relatively short 0-10 day lead time. H06 relates
outflow based on the incoming flow. In this study, only the non-irrigation methodology
from H06 was used to simulate reservoir outflow at each time step ($Q_{out,t}$) since one cannot
assume seasonal irrigation demands will be known globally. Further, the primary purpose
of reservoirs selected in this study is not irrigation. the H06 method estimates outflow as
follows:

$$Q_{out,t} = \begin{cases} k_r Q_{in,t} & (IR = 0.5) \\ (\frac{IR}{0.5})^2 Q_{in,t} + Q_{in,t}\left\{1 - \left(\frac{IR}{0.5}\right)^2\right\} & (0 < IR < 0.5) \end{cases} \qquad (3)$$
where $Q_{in,t}$ is the inflow [m$^3$s$^{-1}$] at time t and $k_r$ is the release coefficient which is
calculated based on Equation 2. The 0.5 threshold value for IR is an empirical condition
derived by Hanasaki et al. (2006).
Unlike H06, D03 relates outflow ($Q_{out,t}$) to current available storage capacity of
the reservoir:
$$Q_{out,t} = \frac{k_{rd}}{\Delta t}(S_t - S_{min})\frac{(S_t - S_{min})}{(S_{max} - S_{min})}^{1.5} \qquad (4)$$
Where Döll empirically derives the release coefficient, $k_{rd}$ = 0.01, $\Delta t$ is the simulation
time step (s), and $S_t$ is the current volume of storage [m$^3$ s$^{-1}$] at time t.  For this study the
D03, $k_{rd}$ was varied usingvalues of 0.01, 0.02, 0.04, 0.06, 0.08, 0.10, 0.20, 0.40, 0.50, 0.60,
0.70, 0.80, and 0.90.
The sensitivity analysis of $k_r$ and $k_{rd}$ can provide useful information on how
coefficients may vary based on geographical and reservoir characteristics such as the
impoundment ratio. The two methods were evaluated and results compared to actual
outflow records provided by the USACE Districts. Two approaches were used to evaluate
model performance: hydrograph assessment of daily and monthly reservoir outflow and
statistical evaluation. The statistical evaluation was performed for daily and monthly
averaged simulated results vs. observations using the Kling-Gupta efficiency (KGE, Gupta
et al., 2009), coefficient of determination (R-Squared), and root mean square error
(RMSE). The KGE value ranges from negative infinity to one. Four levels of performance
were defined for KGE in this study (Tavakoly et al., 2017): poor performance (KGE < 0),
acceptable (0 < KGE < 0.4), good (0.4 < KGE < 0.7), and very good (0.7 < KGE).
Goodness-of-fit values were evaluated to compare simulated discharge to the actual
outflow records provided by the USACE Districts. These are indicators of how well the
models perform. The same goodness-of-fit values are calculated to compare actual
discharge with inflow to assess baseline performance. The baseline condition represents
the treatment of reservoir outflow as naturalized, altogether neglecting reservoir
operations. Thus, the baseline condition is that inflow into the reservoir equals outflow
from the reservoir. To be viable, the reservoir routing scheme should improve results over
the baseline condition in virtually all cases.
A true directly measured daily inflow is not available for most reservoirs, including
those maintained by the USACE. There are two ways that one can acquire a daily reservoir
inflow; estimated using a streamflow model (as in Masaki et al., 2017; Zajac et al., 2017)
or  estimated using a back calculated inflow based on the known discharge and observed
changes in reservoir storage (as in De Vos, 2015). The authors have chosen to utilize a
back calculated inflow because this methodology inherently accounts for all other
withdraws from the reservoir, such as irrigation, evapotranspiration, seepage, etc. This
allows the study to focus exclusively on the reservoir routing methodology.  In fact, that
would double count withdrawals from the reservoir.
2.2.    Study Area
The model evaluations were conducted on 60 reservoirs in the United States
maintained by the U.S. Army Corps of Engineers (USACE).  Figure 1 illustrates reservoirs
used in this study. The primary purpose of 43 of the reservoirs are flood control, six are
hydroelectric, four are recreation, three are water supply, two are classified as other, one is
irrigation, and one is a fish and wildlife pond. Despite most reservoirs in the sample being
primarily purposed as flood control reservoirs, only three of these reservoirs are exclusively
purposed for flood control.  Table 1 describes pertinent characteristics of each reservoir in
this analysis.

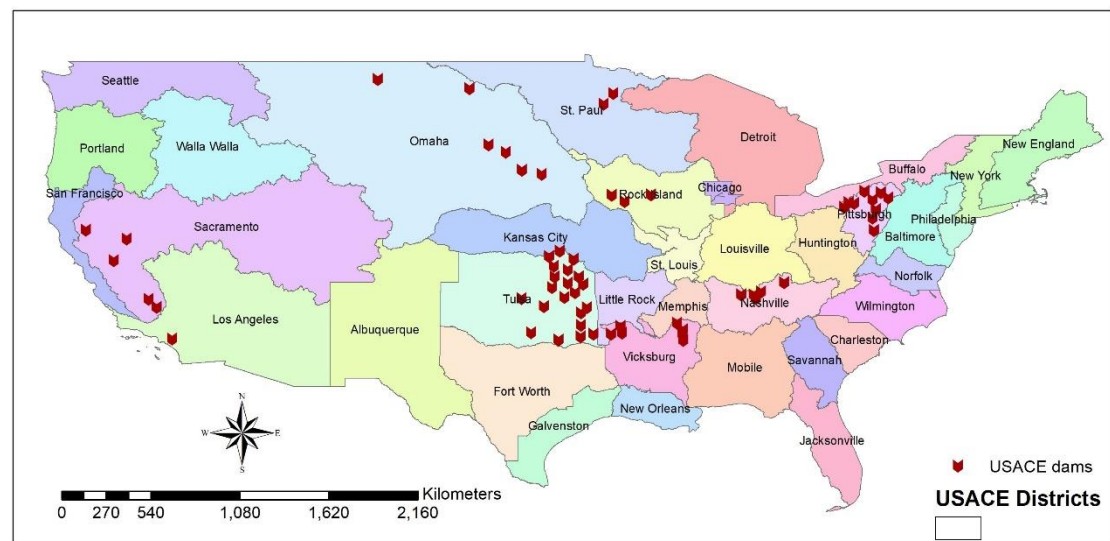

Figure 1. USACE districts and location of reservoirs in this study.

Table 2.  Select statistical characteristics of reservoirs analyzed in this study.

| Characteristic | Range | Mean | Standard Deviation |
|---|---|---|---|
| Minimum Storage (m³ x10⁶) | 0 - 12,377 | 827 | 2,553 |
| Maximum Storage (m³ x10⁶) | 25 - 32,070 | 2,695 | 6,184 |
| Annual Inflow (m³/s) | 0.64 - 780 | 118 | 202 |
| Annual Outflow (m³/s) | 0.66 – 776 | 113 | 195 |
| Impoundment Ratio | 0.03 -15.50 | 1.96 | 2.33 |


# 3.  Results and Discussion

This section describes the overall results of the study.  There is significant
improvement in skill over the baseline (the use of inflow as an estimate of outflow) when
the optimal D03 coefficient is chosen.  D03 tends to outperform the baseline. H06
generally mirrors the results of the baseline. For this reason the discussion largely focuses
on D03.  The authors examine the distribution of best fitting $k_{rd}$ values.  We discuss how
dam systems, annual variability, and simulation time step can influence the ability of D03
to estimate reservoir outflow.  The authors also discuss the potential for numeric
instability in D03 simulations and offer an initial solution to this instability.  We also
provide an overview of the limitations of this study and suggested future work.
3.1.    Overall Model Performances
The  goodness-of-fit  metrics  were  calculated  for  each  reservoir  in  the  study.
Observed inflow is compared with observed outflow to establish a benchmark used to show
whether  implementing  the  two  non-data  driven  reservoir  routing  schemes  improves
estimates for reservoir outflow over the use of unregulated flow as the reservoir outflow
estimate. Figure 2 illustrates the comparison of skill metrics between baseline  and the use
of D03 and H06 to simulate outflow. The KGE, R-Squared, and RMSE for D03 and H06
in Figure 2 represent the best fit results from the sensitivity study. Data points in Figure 2
that fall below the dashed line represent instances where KGE, R-Squared, and RMSE are
lower for the reservoir routing method compared to the baseline. Data points falling above
the dashed line indicate instances where higher KGE, R-Squared, and RMSE were obtained
than the baseline for this study. H06 tends to show minimal utility over the baseline
scenario. In general, H06 does not appear to make outflow estimates worse. Estimates that
have acceptable KGE values in the baseline scenario tend to produce acceptable results
using H06. On the other hand, Figure 2 illustrates that D03 generally tends to increase KGE
and R-Squared, and with this increase in goodness-of-fit, decrease RMSE. Thus, the
general conclusion is that selecting the optimum D03 release coefficient will ultimately
produce an improved estimate of reservoir outflow compared to the baseline. Generally,
H06 will produce an estimated reservoir outflow that performs similarly to the baseline
scenario.

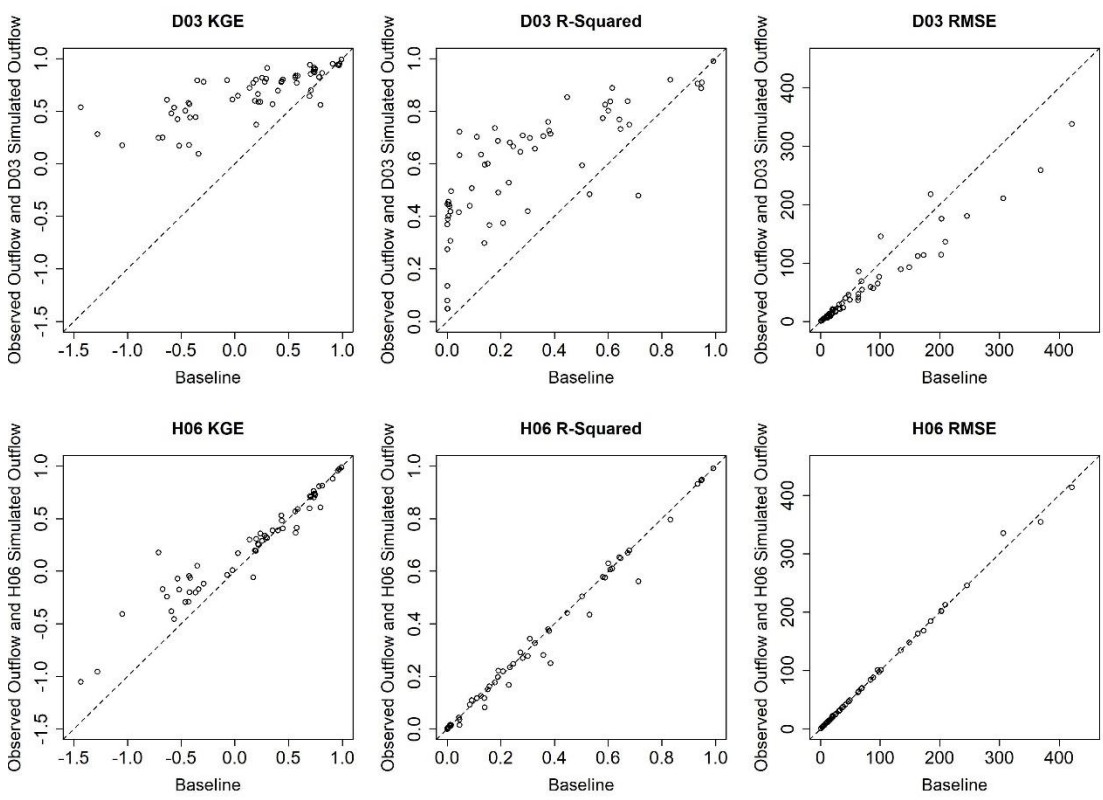

Figure 2. Scatter plots of skill metrics between the use of daily observed inflow as outflow (Baseline) and simulated outflow from best performing D03 and H06 simulations. The dashed line indicates the plane separating increased and decreased skill that results from using either reservoir routing method.

Figure 3 is a geographic representation of the KGE values from the baseline scenario as well as the best performing implementation of the two routing models for each reservoir. In general, D03 outperforms the baseline and H06, particularly in the Tulsa and Pittsburg Districts. H06 tends to provide, at best, minimal improvement in accuracy over the baseline.

D03 tends to improve KGE values at nearly all reservoirs and tends to preserve high KGE values at locations where the baseline is already a good or very good estimator of outflow. Only one of the 60 reservoirs in this study demonstrates a significant reduction in accuracy when D03 is applied. This reservoir, Martis Creek Dam in the Sacramento District, appears to be an outlier in the reservoir sample. Reservoirs with a similar IR and average inflow to Martis Creek Dam and in the same USACE district tended to experience

improvement in model skill with D03.  Overall, when the appropriate $k_{rd}$ value is applied,
D03 improves simulation results over the baseline.

Figure 3a illustrates the wide range of reservoir operating conditions present in the

study. The reservoir dataset contains reservoirs in which the outflow correlates poorly with
the inflow regime as others that correlates well. Figure 3a also portrays significant
geographic clustering where reservoirs in certain regions tend to be less correlated with
inflow and other clusters where observed inflow and observed outflow correlate strongly.
This could indicate that operations at these reservoirs may have a particularly regional
context and may bias towards a particular reservoir routing scheme. However, correlation
between observed inflow and observed outflow and geographic proximity of the reservoirs
does not influence the implementation of either D03 or H06. Thus, the results of this
research indicate no significant geographic constraints in the context of this study.

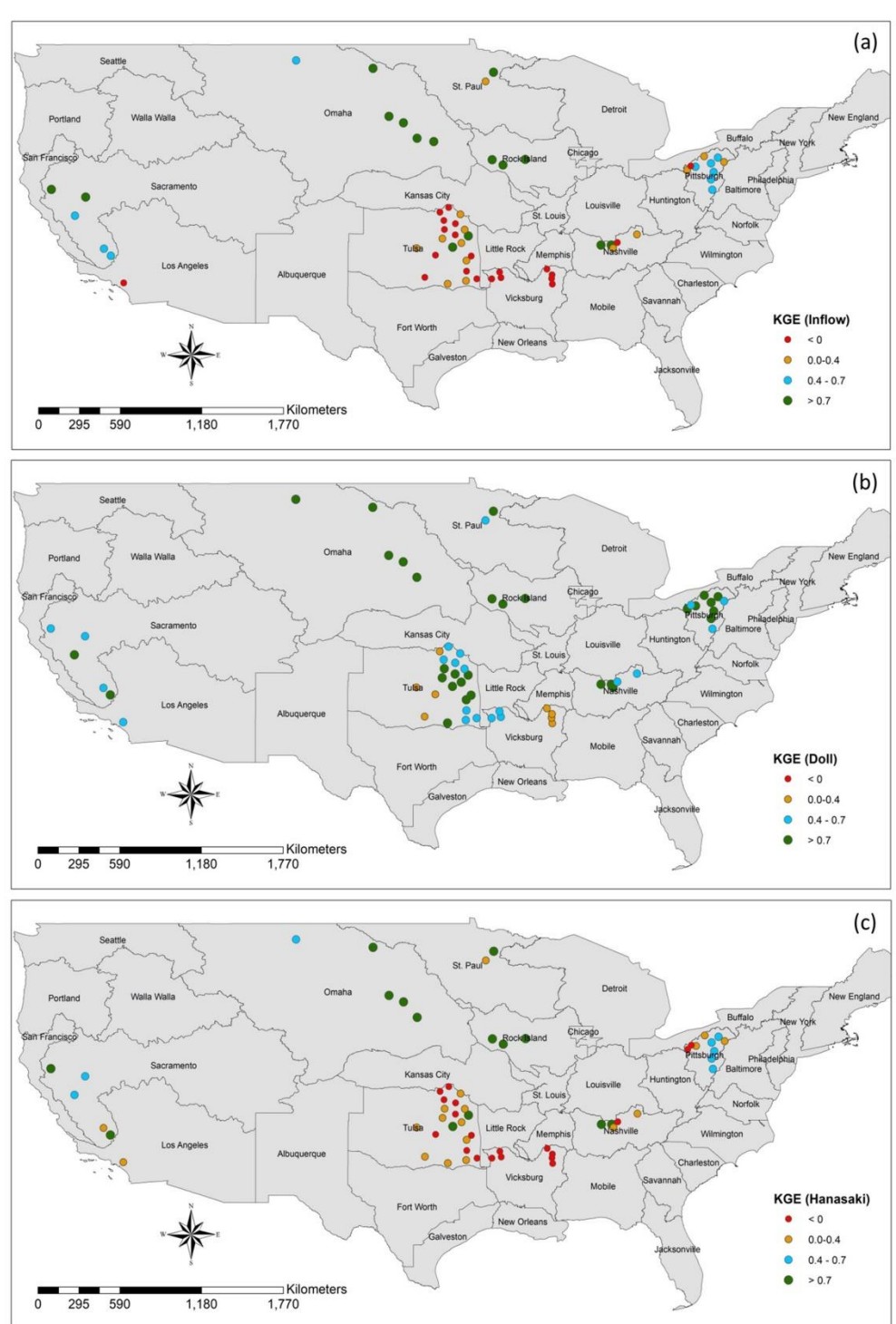


Figure 3. Spatial distribution of KGE comparing observed daily outflow to the each best estimate of outflow:
a) observed inflow b) Döll Method simulated outflow, c) Hanasaki Method simulated outflow for all
reservoirs in this study.  KGE values for the Döll Method and the Hanasaki Method are the maximum KGE
from all coefficient treatments.

Figure 4 presents a proportional bar chart comparing baseline KGE and the highest

KGE value for the range D03 and H06 coefficients.  This plot categorizes KGE
performance using the same bins as Figure 3.  Figure 4 indicates that the best performing
H06 simulation provides only marginal improvement over the baseline condition.
However, the best performing instance of D03 eliminates all poor performing baseline
conditions.  Nearly 87% of all best performing D03 simulations are considered to be good
or very good at accurately capturing reservoir outflows, a 22% increase above the baseline
simulation.

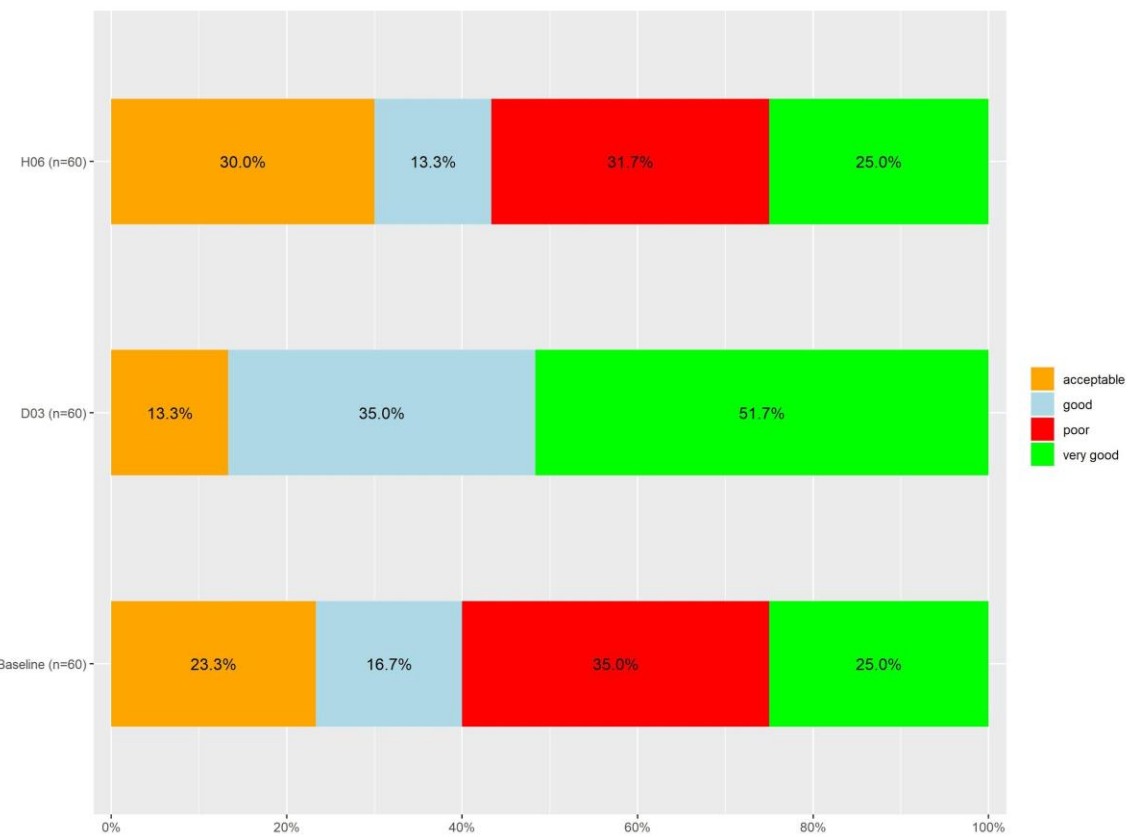


Figure 4. Proportional bar chart comparing the baseline outflow estimation and the best KGE results for D03
and H06.

From multivariate comparison, a negative relationship between two of the best fit

results (KGE and R-Squared) and reservoir IR was found. Figure 5 illustrates this
comparison between IR and each goodness of fit metric for the baseline, D03, and H06.
KGE in particular appears non-linearly correlated to IR. A similar, yet less significant,
negative relationship was found between IR and R-Squared. Little statistical correlation
appears to occur between IR and RMSE. However, KGE and R-Squared values in Figure
5 indicate that the ability to predict outflow using the reservoir routing techniques applied
in this study decreases with reservoir with high IR values.

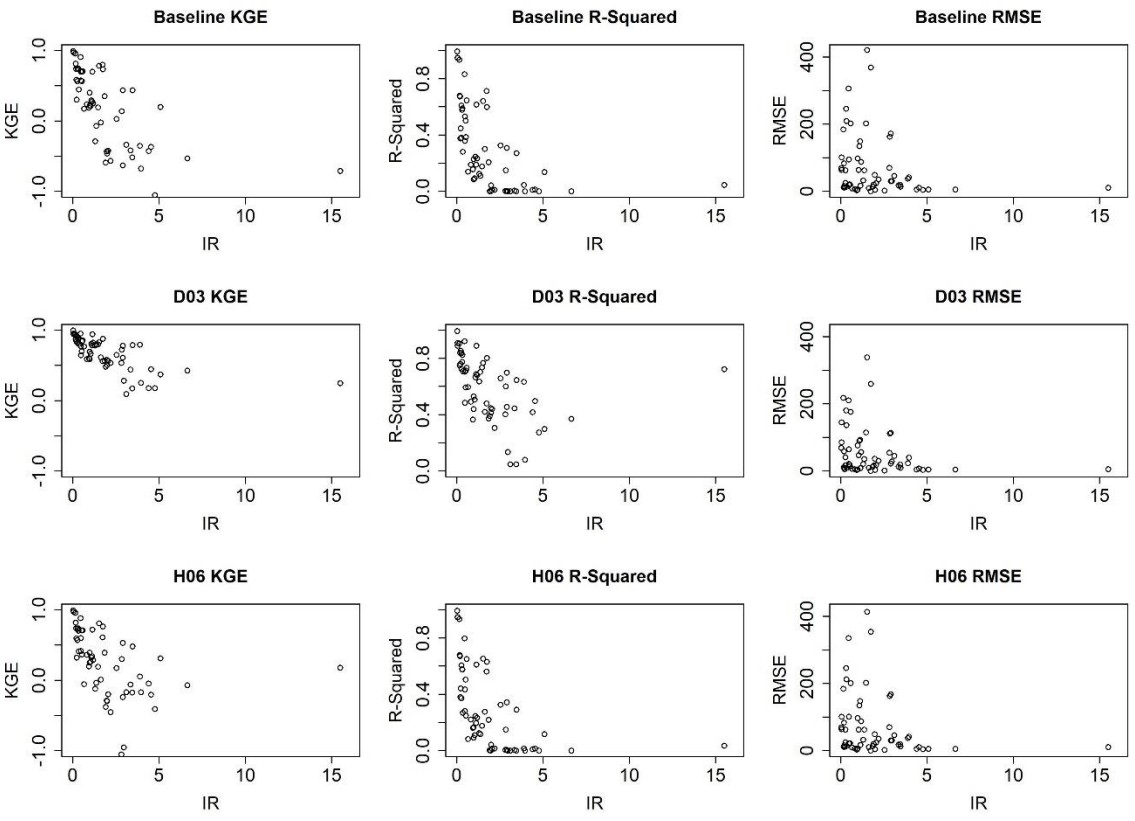


Figure 5. Comparison of IR and best KGE, R-Squared, and RMSE from goodness of fit metrics for baseline,
D03, and H06.
3.2.    Sensitivity Analysis of Models

Because D03 consistently outperforms H06 at daily time steps, D03 was selected

for the sensitivity analysis at daily time steps. The value of $k_{rd}$ coefficient was introduced
as 0.01 in the Döll et al. (2003) study. In this study, $k_{rd}$ values were varied to obtain
maximum KGE and R-Squared and minimum RMSE.  Figure 6 demonstrates the
dispersion of $k_{rd}$ values which maximize the model skill for all reservoirs in this study.
For all model skill metrics, $k_{rd}$=0.90 tends to be the most prevalent $k_{rd}$ value that
maximizes model skill. In only two of the 60 reservoirs (Sardis Dam and Enid Dam) $k_{rd} =$
0.01 maximizes R-Squared and minimizes RMSE for the range of $k_{rd}$ coefficients. This
research suggests that the $k_{rd} = 0.01$ is not necessarily the optimum coefficient to
maximize model performance using a daily simulation time step.

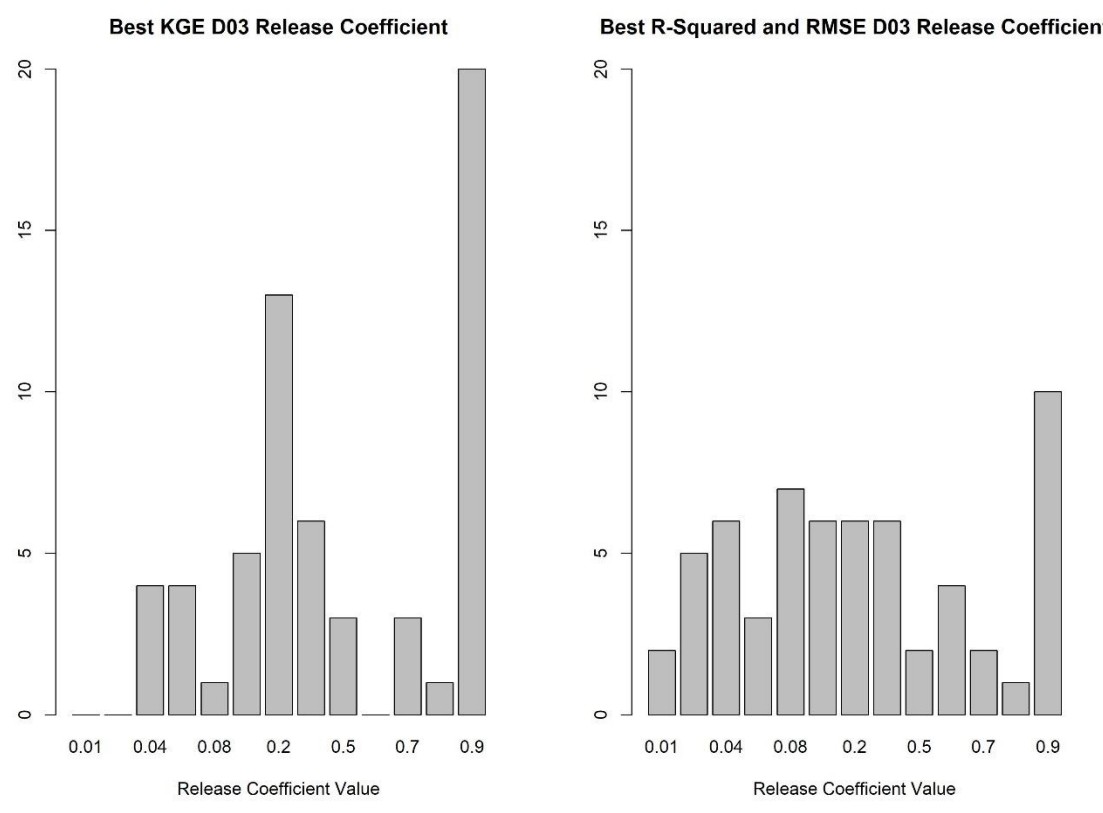


Figure 6. Bar charts of $k_{rd}$ values that maximize KGE and correlation and minimize RMSE.

Investigating the linkage between dam characteristics and the best performing $k_{rd}$

yields no clear relationship. Evaluation of correlation between IR, coefficient of variation
of inflow, ratio of average inflow to average outflow, and geographic location shows low
correlation between each variable and best performing $k_{rd}$ value. However, the range of
best performing $k_{rd}$ within this analysis and as demonstrated in Figure 6 suggests that the
value is not constant across all reservoirs. Thus, as one implements D03 within their
hydrologic forecasting framework, $k_{rd}$ may be adjusted to optimize streamflow estimates
to gage observations, like those curated by the Global Runoff Data Centre (GRDC, 2018),
when available.
3.3.    Dam Systems and Reservoir Routing
Reservoirs in the Vicksburg and Omaha districts were selected to evaluate
performance of D03 in environments where reservoirs operate in a coordinated fashion.
We broadly refer to these as dam systems. The case of the Vicksburg and Omaha district
reservoirs highlights two distinct types of dam systems; one where the dams do not
contribute inflow into one another but still coordinate their releases (in parallel) and another
where upstream releases flow into downstream reservoirs (in series).
A subset of the reservoirs in the Vicksburg District comprise the Yazoo Basin
Headwaters Project. Although the reservoirs in the Yazoo Basin Headwaters Project are
not directly connected, the reservoir operators coordinate operations in order to minimize
flooding in Mississippi's Delta region (Arkabutla Lake History, 2017; USACE, 1987). The
operation of these reservoirs presents an interesting case in which the non-date driven
models in this study do not characterize the nature of the dam releases well. The modeled
results at four Vicksburg District dams yield only minimal improvement over unregulated
(i.e. naturalized) flow at these reservoirs. The decrease in reservoir routing performance
can be attributed to the large impoundment ratios at these dams indicating the reservoir
storage is large relative to annual volume of inflow.
The reservoirs of interest in the Vicksburg District include Arkabutla, Sardis, Enid,
and Grenada. These dams function in parallel on tributaries of the lower Mississippi River,
namely the Coldwater River, Little Tallahatchie River, Yocona River, and Yalobusha
River, respectively. Together, these dams control flooding in northern Mississippi as part
of the Yazoo Basin Headwaters Project (Arkabutla Lake History, 2017; USACE, 1987).
The Yazoo Basin reservoirs discharge directly into the heavily regulated Mississippi River
(Meade and Moody, 2010). The reservoirs operate to ensure high releases are not
concurrent with large flows upstream on the Mississippi to avoid devastating flooding to
the low-lying Louisiana delta regions. This requires a high level of coordination throughout
the Yazoo Basin Headwater Project and with regulation upstream on the Mississippi.
Additionally, each of the Yazoo Basin reservoirs have a substantial impoundment ratio,
ranging from 2.96-3.95. In other words, the reservoirs are capable of containing large
volumes of water to mitigate downstream impacts. Thus, current pool levels and forecasted
inflow at these four reservoirs do not substantially influence release decisions. The
reservoirs also have the capacity to absorb large flood events. As a result, they do not seem
to follow the same functional form as the majority of dams in this study.
Figure 7 from Sardis Dam in the Yazoo Basin Headwaters Project demonstrates the
hydrograph comparing observed inflow and outflow and the modeled outflow that provides
the highest KGE (D03, $k_{rd}$=0.90) for the year 2008. Figure 7 demonstrates that peak
outflows do not tend to correspond to the time at which peak inflow occurs. In fact, release
rates at Sardis Dam are at a minimum during the peak inflow time period.  This pattern
repeats at each of the reservoirs in the Yazoo Basin Headwaters Project indicating that
inflow and consumed storage are not substantial predictors of outflow timing at these
reservoirs.  This exemplifies the lack of correlation between observed inflow and observed
outflow at reservoirs within the Yazoo Basin Headwaters Project.

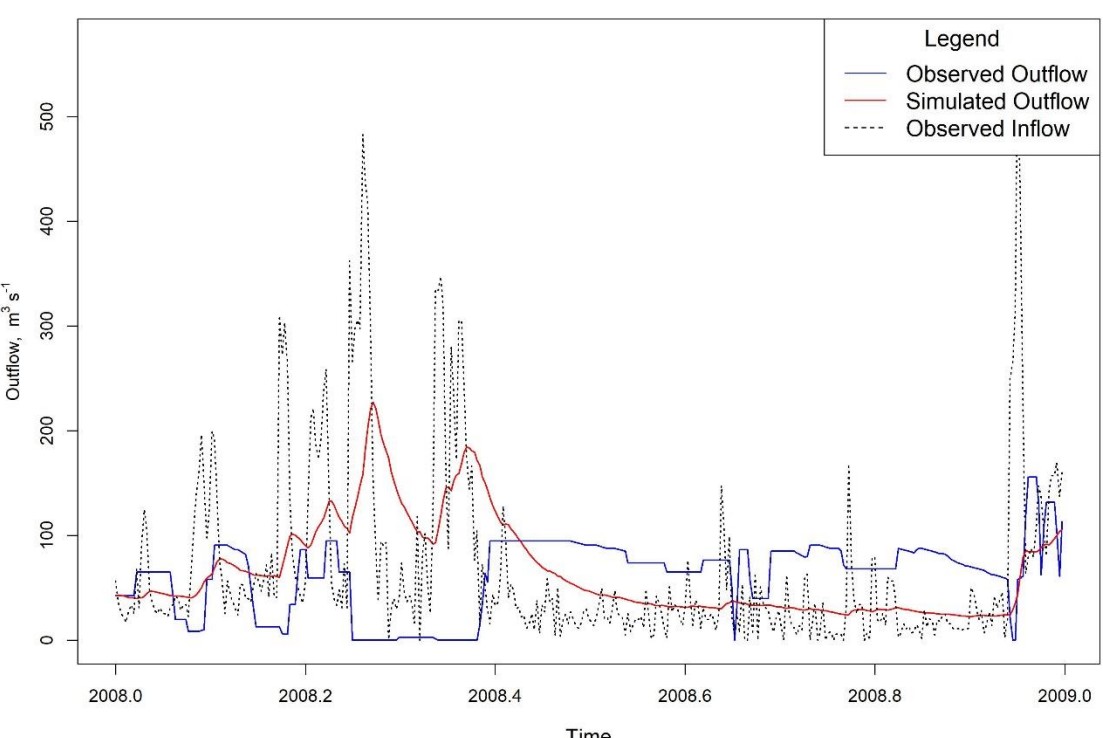


Figure 7. Hydrographs of observed inflow and outflow versus simulated outflow with the highest KGE value
at Sardis Dam (Döll method $k_{rd}$=0.90). KGE comparing observed Inflow and outflow = - 0.34; KGE
comparing simulated and observed outflows= 0.095
Dams operating in series represent a specific case where compounding model error
is a particular concern. USACE operates several large dams in series on the Missouri River.
These include Fort Peck, Garrison, Oahe, Big Bend, Fort Randall, and Gavins Point within
in the Omaha District (Lund and Ferreira, 1996). For this cascading system on the Missouri
River, inflow appears to be a progressively stronger predictor of outflow from upstream to
downstream. At the upstream end the baseline yielded a KGE=0.43 at Fork Peck with a
KGE=0.99 downstream at Gavins Point Dam. Figure 8 provides a comparison of observed
inflow and outflow along with simulated outflow for Gavins Point Dam. D03 tends to
provide a slightly better estimate of outflow compared with inflow, except in the instance
of Big Bend Dam. At Big Bend Dam, H06 produces an estimate of outflow more consistent
with observed outflow than either D03 or inflow alone. However, the differences are almost
trivial considering how well inflow alone performed in this case. D03 is particularly
accurate during peak inflow conditions, for example the large hydrologic event in mid-
2011 at Gavins Point Dam in Figure 8. The performance of non-data driven approaches in
this instance is promising since compounding errors are a large concern in this type of
system. Other instances involving dams in series should be evaluated to determine out if
these findings hold more generally.

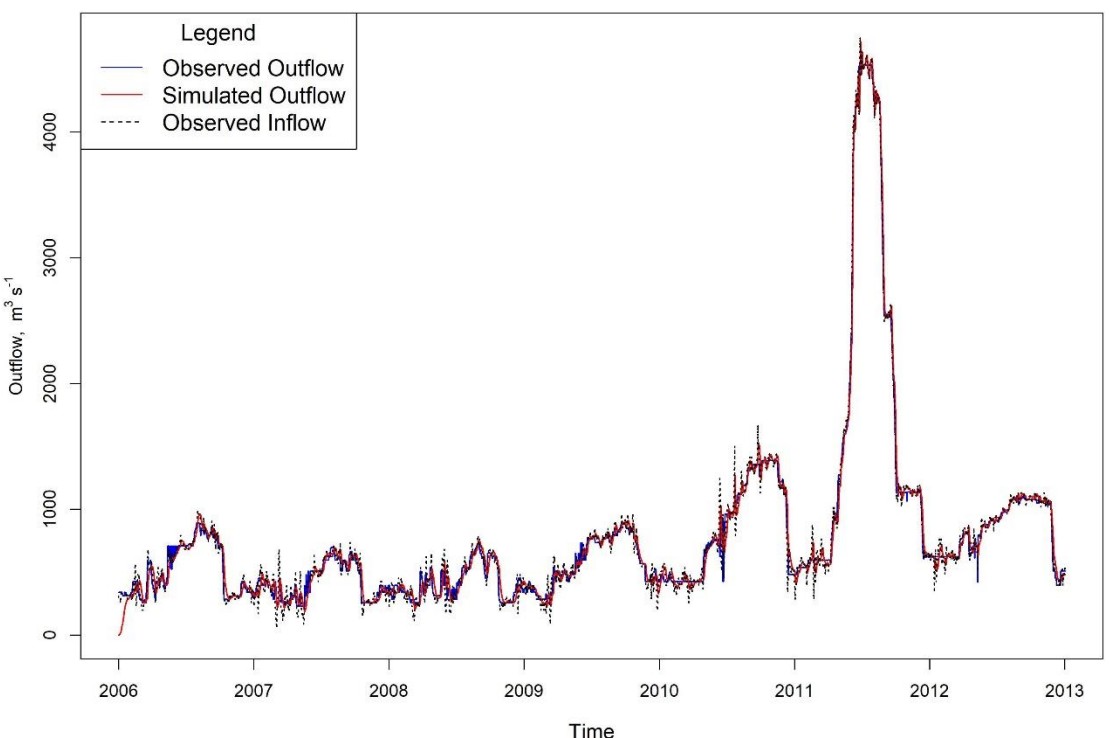


Figure 8. Hydrographs of observed inflow and outflow versus simulated outflow with the highest KGE value
at Gavins Point Dam (Döll method $k_{rd}$ =0.04). KGE comparing observed Inflow and outflow = 0.99; KGE
comparing simulated and observed outflows= 0.99.

Reservoir management is unique in both the Yazoo Basin Headwaters Project and

the Missouri River. The operators of dams within the Yazoo Basin Headwaters Project tend
to regulate outflow in a manner that is more in line with downstream conditions. The
attention to downstream conditions is due mainly to the impact that downstream floods will
have on the low-lying communities within the Louisiana Delta. The dams in the Yazoo
Basin Headwaters Project have among the highest impoundment ratios, which inherently
reduces the influence of upstream conditions in discharge decisions. The non-data driven
approaches evaluated here do not account for downstream conditions and thus do not
perform well in this instance, particularly where large impoundment ratios allow operators
considerable leeway.

On the other hand, the non-data driven approaches tend to perform well when

inflow conditions dictate discharge decisions as we see on the Missouri River system.
Reservoirs with smaller impoundment ratios are naturally more responsive to inflow
requiring greater consideration for upstream conditions. D03 showed relatively small
improvement of outflow estimates compared to inflow as a predictor of outflow in the
Yazoo Basin Reservoirs, while the method provided reasonable estimates in dam systems
like the Missouri River system. Therefore, it can be inferred that D03 is more applicable
for dam systems where reservoir management focuses on upstream hydrologic conditions,
while large impoundment ratios may be indicative of reservoirs where downstream
conditions are more likely to prevail. This would likely apply for H06 as well since that
method links outflow to inflow more directly.
3.4.    Wet and Dry Year Comparison

Figure 8 shows results for wet and dry years at two reservoirs considered to be

representative of this study. D03 provides a relatively good estimate of outflow at Union
City Dam (Pittsburg District) in Figure 9a and Figure 9c.  D03 performs relatively poorly
at Arcadia Lake (Tulsa District) in Figure 9b and Figure 9d. In the case of Union City Dam,
D03 tends to produce a noticeable improvement in model skill during both a relatively wet
year and a relatively dry year. The performance (Figure 9a and Figure 9c) seems to be
independent of wet or dry conditions, at least on an annual basis. This does not hold for
Arcadia Lake. The model shows modest skill at Arcadia Lake during the wet year (Figure
9b), but almost none during the dry year.

There appears to be a difference in the timing discharges between at the two

locations in Figure 9. D03 appears to estimate the right amount of volume released during
the wet year at Arcadia Lake (Figure 9b).  However, the timing of the observed release is
delayed until a relatively dry period begins. The lag could indicate that water is being
retained, possibly for use in irrigation or domestic supply. In this instance, Arcadia Lake
supplies water to the city of Edmond, Oklahoma which may influence release decisions
(Arcadia Lake, 2020).

D03 performs much more poorly during the 2006 dry year at Arcadia Lake (Figure

9d). The model does not predict the sporadic releases throughout the year. The inflow
events in that year are not substantial enough to affect storage meaningfully, thus we see
almost no response in the modeled output. Observed outflows demonstrate that beyond two
relatively high-volume reservoir releases during 2006, the reservoir releases are restricted
to practically no outflow the rest of the year. D03 does not anticipate the two large releases,
as the reservoir storage does not dramatically shift in either instance. D03 estimates a near
constant discharge over the entire year with almost no storage change.

Results for wet years and dry years appear to be fairly mixed. Indications are that

the performance of D03 could be somewhat site specific. However, reservoirs that tend to
be less responsive to storage fluctuations are not represented well in D03 since storage
fluctuations drive the model. Arcadia Lake has an IR of about 4.75 which is relatively high.
Union City Dam has an IR of about 0.24, which is relatively low. IR is a good indicator of
reservoir responsiveness to storage fluctuations.  A lack of reservoir responsiveness to
storage fluctuations could result in two different types of error when D03 is implemented
within a large-spatial-scale hydrologic model. First, forecasted outflow could easily
mistime a hydrologic event, particularly during wet years, as Figure 9b demonstrates.
Second, the authors anticipate that if the storage does not dramatically fluctuate during a
dry year the estimated reservoir release will not anticipate sporadic releases for irrigation
and other purposeful discharges. Unaccounted for, these large but short duration releases
may lead to a consistent overestimation of reservoir outflow for the entire dry year period.

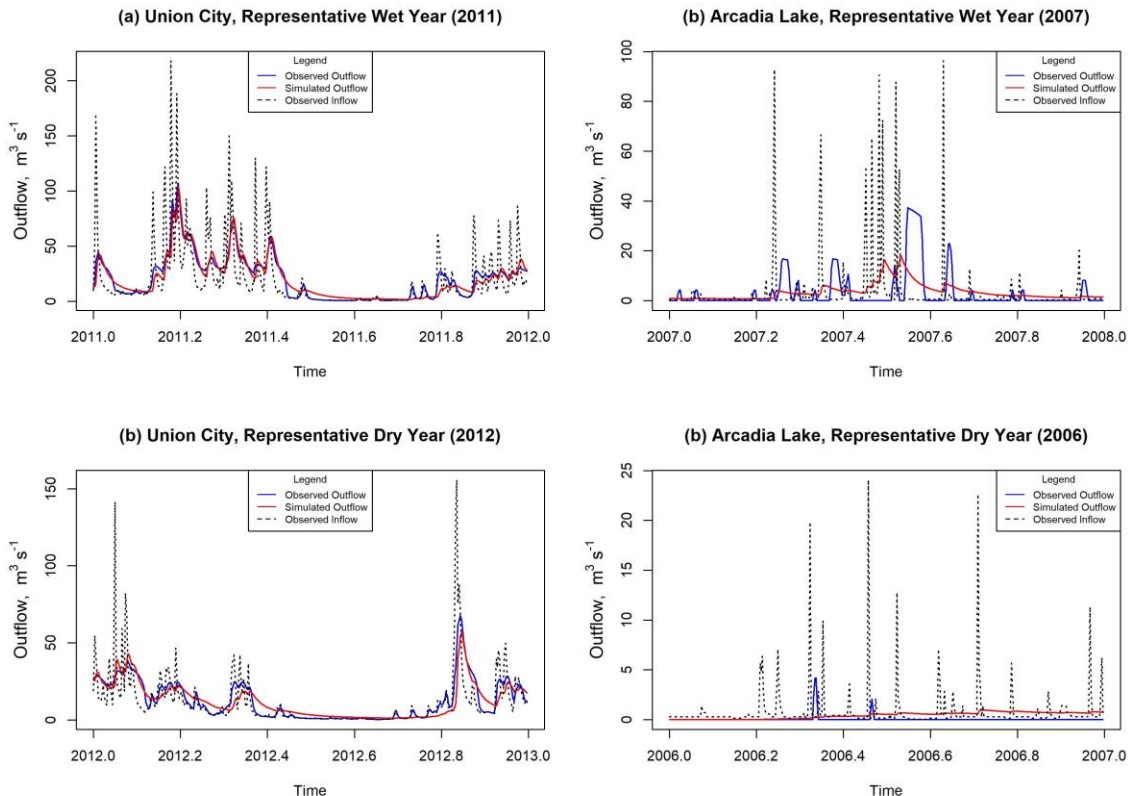


Figure 9. Two reservoirs where D03 tends to perform very good and poor: outflow: a) wet year Union City
Dam 2011; b) wet year Arcadia Lake 2007; c) dry year Union City Dam 2012; and d) dry year Arcadia
Lake 2006.

## 3.5. Effects of Time Step on Model Performance

Model comparisons are conducted for daily and monthly time steps. Table 2

illustrates the results at Fort Peck, Garrison Dam, Oahe Dam, and Fort Randall Dam, each

of which appears in the Hanasaki et al. (2006) study and this research. Table 2 also contains

Sardis Dam, Mosquito Creek Dam, and Prado Dam, which are not included in Hanasaki et

al. (2006). Results illustrate that the time scale at which comparisons are conducted can

influence simulation results. The monthly comparison amongst Fort Peck, Garrison, Oahe,

and Fort Randall is in agreement with the conclusions of Hanasaki et al. (2006). However,

when the simulation time step changes to a daily time step, the skill of H06 and D03 reverse

and D03 tends to outperform H06. In additional reservoirs (Sardis and Prado), the results

indicate that D03 outperformed H06 at both daily and monthly time steps, based upon
KGE. However, the results at Mosquito Creek reservoir tend to follow the original
Hanasaki et al. (2006) results.

The time-scale effect upon model performance may relate to how well observed

inflow correlates with observed outflow. Examining Table 2, H06 outperforms D03 when
observed inflow and observed outflow are relatively well correlated. The effect is nullified
when the inverse is true. H06 estimates outflow as a ratio of inflow, which may be a better
estimate of outflow at the monthly time scale, particularly when discharge tracks closely
with inflow. However, H06 will fluctuate at the smaller time steps due to inherent
variations in inflow. D03 tends to vary less at a daily time step and may be a better estimate
of outflow at sub-monthly time steps.

The hydrographs from Fort Randall Dam further illustrate the relationships between

time step and model skill, particularly during high flow events. Daily and monthly
comparisons between observation and simulations for Fort Randall Dam are shown in
Figure 10. Figure 10 compares the daily and monthly simulations with observations. Figure
10a shows that the H06 simulations perform better than D03 for monthly time steps,
particularly during the high inflow periods in 2011.  D03 tends to overestimate reservoir
outflow, while H06 correlates well with inflow and better matches the peak flow of 2011.
At a diurnal time step (Figure 10b), H06 tends to be hypersensitive to inflow variations and
overestimates outflow, whereas D03 provides a better approximation of outflow during the
2011 high flow event at a daily time step.


Table 3. Comparison of daily and monthly KGE values at selected reservoirs. The $\alpha$ and $k_{rd}$ values
represent the highest KGE values for Hanasaki and Döll methods respectively.

| Reservoir | Daily KGE | | | Monthly KGE | | |
|---|---|---|---|---|---|---|
| | Inflow | Hanasaki | Döll | Inflow | Hanasaki | Döll |
| **Fort Peck** $\alpha$=0.95 $k_{rd}$=0.04 | 0.43 | 0.53 | 0.78 | 0.54 | 0.62 | 0.51 |
| **Garrison Dam** $\alpha$=0.95 $k_{rd}$=0.06 | 0.73 | 0.76 | 0.88 | 0.78 | 0.80 | 0.59 |
| **Oahe Dam** $\alpha$=0.95 $k_{rd}$=0.20 | 0.78 | 0.81 | 0.83 | 0.84 | 0.86 | 0.76 |
| **Fort Randall Dam** $\alpha$=0.95 $k_{rd}$=0.20 | 0.91 | 0.88 | 0.95 | 0.96 | 0.93 | 0.67 |
| **Sardis Dam** $\alpha$=0.95 $k_{rd}$=0.90 | -0.34 | -0.17 | 0.09 | 0.06 | -0.03 | 0.16 |
| **Mosquito Creek Dam** $\alpha$=0.45 $k_{rd}$=0.70 | -0.46 | -0.29 | 0.51 | 0.49 | 0.60 | 0.39 |
| **Prado Dam** $\alpha$=0.95 $k_{rd}$=0.50 | -0.02 | 0.01 | 0.61 | 0.32 | 0.61 | 0.71 |


It is possible that the conclusions of Hanasaki et al. (2006) suggesting better performance
of H06 at the monthly-scale depend on how closely discharge from the dam tracks inflow.
D03 may be a better candidate for integration into daily flow forecasting models.

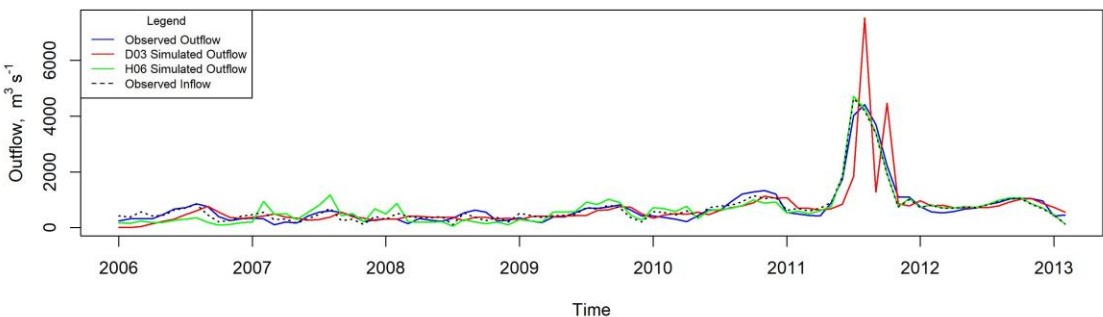

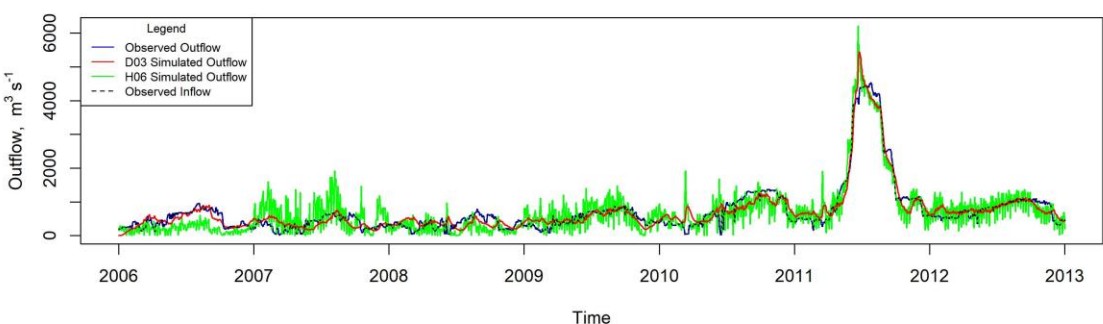

Figure 10. Comparison of simulated outflow for the Fort Randall Dam with Hanasaki and Döll methods for (a) monthly and (b) daily time steps.

3.6.    Model Stability

Although D03 outperformed H06 when using a daily time step, D03 demonstrated

some instability for high $k_{rd}$ values. This instability occurs at three reservoirs in this study.

The cause of the instability is a combination of a reservoir having a low IR and a sharp

change in the inflow to a reservoir. For instance, inflow into Old Hickory Dam in the

Nashville District (IR = 0.04) increased by roughly two orders of magnitude in a matter of

a few days in May 2010. During this event, the available storage filled up, necessitating a

substantial increase in release flow to prevent overtopping.  This occurred within a single

time step in the model (D03) and the outflow responded in kind in the next subsequent time

step which then drained the reservoir below the specified minimum storage resulting in a
non-computable imaginary number as the next solution.
Several solutions are posited to address D03 instability. One solution could be to
varying $k_{rd}$ values dynamically to mimic reservoir behavior. During large hydrologic
events the value of $k_{rd}$ could reduce the peak of the outflow hydrograph, and then increase
during normal events. Another solution is the inclusion of rules and an expanded system
of equations that govern the solution. Because the intention of D03 is to approximate flow
at a free-flowing weir, coupling operational rules with the simulation may better
approximate reality. The rules may be as simple as switching behavior or the algorithm
when storage approaches either minimum or maximum reservoir storage. A simple
condition was tested for when storage drops below the minimum storage during the daily
time step:

$$if\ S_t \le S_{min} \Rightarrow \begin{cases} S_t = S_{min} \\ Q_{out} = Q_{in} + \dfrac{S_t - S_{min}}{\Delta t} \end{cases}$$

(5)

This condition prevents the reservoir from falling below the minimum storage. Outflow
from Old Hickory Dam was re-simulated with $k_{rd} = 0.9$ and the new minimum storage
condition (Equation 5). The proposed modification resulted in simulated outflow shown in
Figure 11. Outflow is substantially overestimated for one-time step and drops to zero at the
next time step. While an oversimplification of actual operations, this condition is similar
to an emergency spillway discharge to prevent overtopping. The dam releases tremendous
flow for a brief period, when the maximum storage is nearly exceeded and then inhibits the
discharge when the storage is at the minimum capacity. The benefit of this modification is
that additional reservoir information is not required. However, further testing and
evaluation should be performed to validate this refinement.

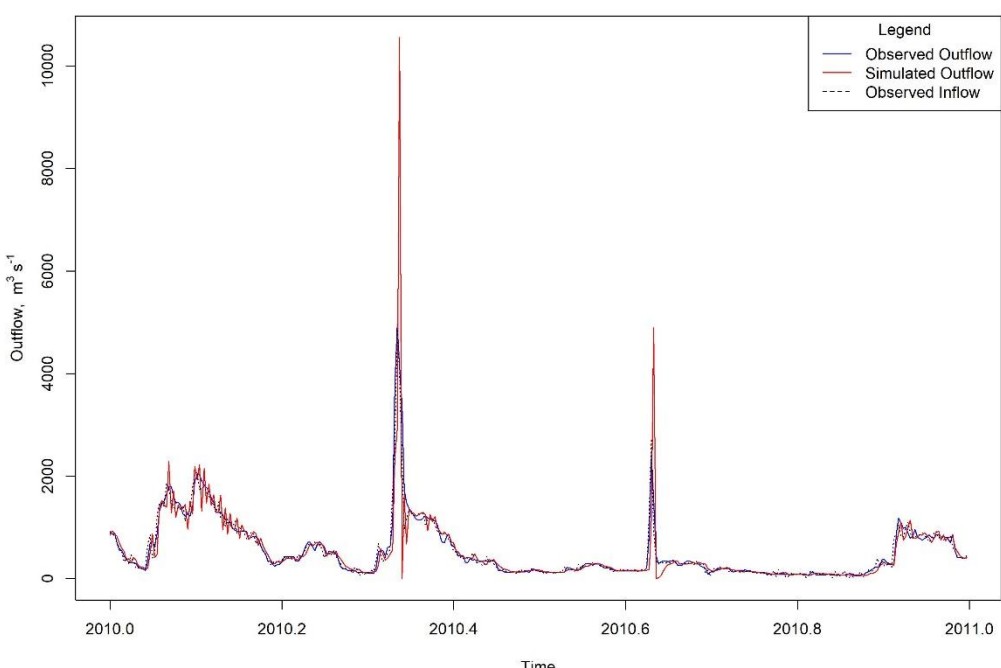


Figure 11. Outflow simulation for the Old Hickory Dam using the proposed modification of the Doll method
for $k_{rd}$=0.4.
3.7.    Limitations

The available sample of dams for this study has some inherent limitations. The vast

majority of reservoirs in the sample are primarily purposed as flood control reservoirs with
various secondary purposes. They are all commonly operated by USACE. And the dams
function within a predominately temperate climate across the United States.    These
limitations preclude assertions regarding the effect the operating objective, dam ownership,
or country of operation on reservoir routing performance.

The abbreviated length of the historical records presents another limitation. The

evaluation period is limited to a six-year window which may not account for the total range

of operational environments for each dam.  Thus, this evaluation likely does not capture

and evaluate D03 and H06 under absolute extreme circumstances.

All inflow utilized in this study is back calculated from observed changes in storage

and known discharges. This indirect method can lead to negative inflow values when losses

due to seepage, evapotranspiration, or other types of withdrawals are underestimated. De

Vos (2015) also noted that they used back-calculated inflow in their study. It is unclear

whether Hanasaki et al. (2006) made use of direct observations, but it is worth noting that

direct observations of total reservoir inflow are not readily available in most cases.

This study is limited to models that only require inputs related to reservoir inflow and

storage, primarily to provide insight into the reliability of these measures as indicators of

reservoir outflow. Because this study utilizes a back calculated reservoir inflow, inclusion

of reservoir withdrawal would also lead to an overestimation of water withdrawals from

the reservoir.  Both D03 and H06 can account for withdrawals but because of the focus of

this study and the data utilized, the authors do not pursue an estimation of reservoir

withdrawal in this study. Thus, we have not included more sophisticated approaches,

such as Burek et al. (2013) or Zhao et al. (2016) within this study. Beyond this study of

sensitivity analysis, no formal calibration procedure was undertaken. A formal calibration

of $k_{rd}$ in both D03 and H06 would be better suited for the insertion of the reservoir

routing scheme within a hydrologic routing scheme.  This study is investigating the

feasibility of these methods in 0-10 day lead time, diurnal forecasting and is a precursor

to implementation in hydrologic routing schemes.  There is limited benefit to standalone

calibration of the $k_{rd}$ coefficients, given that reservoir outflow information is rarely
available at global scales. Operational calibration of $k_{rd}$ would be challenging without
reservoir release records. Zajac et al. (2017) discuss the need for an open access database
of daily reservoir records, but no such database is known to be available at this time.
Thus, this study does not undertake any standalone, formal calibration of $k_{rd}$.
3.8.    Future Work

D03 consistently improved simulated, daily streamflow estimates over naturalized

flow conditions in the selected reservoirs of this study, suggesting that D03 can potentially
improve global streamflow forecasting that do not already account for lakes and reservoirs.
D03 performed particularly well at daily time steps commensurate with many large-scale
stream routing models. The incorporation of D03 and H06 can be considered as modules
in large-scale river routing models such as Routing Application for Parallel computatIon
of Discharge (RAPID, David et al., 2011). The RAPID model is a river routing model that
can simultanusley compute streamflow in river networks with thousands of river reaches.
This will enable widespread testing and evaluation over large hydrologically diverse areas.

The research presented in this article should guide a number of follow-up

evaluations that will broaden the scope of this evaluation.

•   We determined that $k_{rd}$ can be varied to improve performance but have no

guidance on how to relate $k_{rd}$ to a given reservoir. Future studies should

determine how to assign release coefficients to reservoirs.

•   We have chosen parsimonious approaches that minimize assumptions. We

have not compared D03 or H06 to more complex models such as Burek et

al. (2013) or Zhao et al. (2016) which require these assumptions. Future

work will examine tradeoffs between model complexity and performance.

• Insertion of D03 into large-scale river routing models can facilitate studies

of how their results influence overall hydrologic performance, particularly

at locations downstream of reservoirs.

• Three quarters of the sampled dams have their primary purpose for flood

control. Efforts to fill the existing dataset with reservoirs that are primarily

irrigation, water supply, hydroelectric, recreation, and fish and wildlife

habitat and analyze the impacts of use on model performance should be

undertaken.

• The non-data-driven methods considered are conceptualizations of

reservoir operations that can be adapted to utilize remotely sensed

information, much like the data-driven methods previously mentioned.

Non-data-driven methods can be linked to statistical fitting techniques, but

they are capable of being employed independent of such pairings. However,

the non-data-driven reservoir routing schemes could be enhanced by

assimilating remotely sensed data, e.g. near real-time changes in storage

resolved from satellite altimetry, and eventually the planned NASA Surface

Water and Ocean Topography (SWOT) Mission. This information could

constrain reservoir simulations to improve global streamflow forecasts

(Yoon and Beighley, 2015).

• Because D03 skill tends to decline with increases in IR, an over-year

simulation capability similar to that proposed by De Vos (2015) may allow

for a better means of simulating diurnal reservoirs from reservoirs with large
IR.  Over-year reservoirs have high IRs and yearly cycles of water storage
and release are not necessary (Adeloye and Montaseri, 2000; Vogel et al.,

1999).

# 4.  Conclusions

This research compares two parsimonious reservoir routing methods (D03 and H06)
with the intent to determine if these methods can be effective at estimating diurnal reservoir
outflow in diurnal, medium-range streamflow forecasting.  These methods were compared
across 60 USACE operated reservoirs at a daily time step. Results show that D03 tends to
outperform H06 at a daily time step. An in depth examination of these results yields the
following conclusions.
• The complexity and data requirements of both D03 and H06 are low and thus

computationally inexpensive. Both can be feasibly implemented at large spatial

scales at a daily or sub-daily time step.

• When the best performing $k_{rd}$ is implemented within D03 we find a substantial

improvement in the model skill over the baseline for nearly all reservoirs at a

daily time step.  H06 offers only a minimal improvement over the baseline when

the best $k_{rd}$ is implemented for a daily time step.  For the categories of KGE

specified (Tavakoly et al., 2017), the best performing D03 eliminates all poor

performing baseline conditions and increases the proportion of good or very

good performing sites by 22%.

• There is a statistical relationship between reservoir IR and two of the skill

metrics applied (KGE and R-Squared). Given that reservoirs with high IR

typically are less responsive to short-term fluctuations in inflow and storage,

the correlation between these variables is plausible. Further investigation of

dam characteristics, such as if the dams operate in series or in parallel and wet

and dry year considerations are further evidence of the correlation between the

IR and D03 and H06 skill.

• Simulation time step appears to be an important component in reservoir routing

skill. The comparison of the two methods by Hanasaki et al. (2006) are based

on monthly reservoir outflows and conclusions may not hold within diurnal

forecasting schemes. At overlapping locations, this study replicates the results

reported by Hanasaki et al. for monthly time steps. However, the Hamasaki et

al. findings do not hold for a daily time step evaluation.

• The best value for the empirical Döll coefficient, $k_{rd}$, can vary. Optimal values

were typically greater than the $k_{rd}$=0.01 value which Döll et al. (2003) derived.

This suggests that $k_{rd}$ could be a potential calibration parameter within a large-

scale hydrologic modeling framework much like a weir coefficient, which is

specific to a particular type of weir.

• The Yazoo Basin Headwaters Project (Arkabutla Lake History, 2017; USACE,

1987) is an interesting case study in how reservoir system complexity can be

difficult to model. The Yazoo Basin Headwaters Project considers downstream

flow conditions as the dominant criteria in dam operation. Thus, the inflow and

available storage volume are poor predictors for determining reservoir

discharge in this type of management scheme. D03 appeared to scale flow

correctly at these reservoirs and improve reservoir overall skill, but timing of

the releases is not well represented and thus skill improvement is only minimal.

• Dam discharges in the Missouri River Reservoir System (Lund and Ferreira,

1996) are more correlated with storage volume and inflow conditions, which

lends itself to the two non-data-driven approaches evaluated here. D03 is

particularly capable of accurately modeling daily reservoir outflows in reservoir

systems that correlate well with storage and inflow fluctuations. Concerns

related to model error being compounded through a series dams may be

mitigated somewhat by the fact that inflow appears to be a progressively

stronger predictor of outflow further downstream in these types of systems.

• Numerical stability of D03 is a concern, particularly with higher $k_{rd}$ values.

These stability concerns originate at reservoirs with small active storage

capacity during high inflow events. Additional model refinement can overcome

these stability concerns.

• D03 showed minimal bias during relatively wet and dry years. Timing of

releases can be influenced by wet years and the magnitude appears to be

affected during dry years. D03 appears to be most applicable for dam systems

where reservoir management focuses on upstream hydrologic conditions. Large

IRs could indicate reservoirs where downstream conditions are more likely to

influence release decisions at the reservoir.

## 5.  Data Availability

All input, output, and evaluation data compiled for this study are available on request for scientific purposes from Joseph Gutenson (jlgutenson@gmail.com) or Ahmad Tavakoly (ahmad.a.tavakoly@erdc.dren.mil).

## 6.  Author Contributions

JLG developed the research questions, analyzed the data, and compiled the paper.  AAT compiled the datasets, developed the research questions, took part in critical discussion of the paper, and reviewed the paper.  MDW provided input for the formulation of the research questions, took part in critical discussion of the paper, and reviewed the paper. MLF provided input for the formulation of the research questions, took part in critical discussion of the paper, and reviewed the paper.

## 7.  Competing Interests

The authors declare that they have no conflict of interest.

## 8.  Acknowledgments

This project was partially funded by the Deputy Assistant Secretary of the Army for Research and Technology through the U.S. Army Engineer Research and Development Center (ERDC) Military Engineering (ME) work package entitled Austere Entry and the Geospatial Research and Engineering (GRE), Environmental Quality and Installations (EQI) work package entitled Climate Adaptive Mission Planning (CAMP), and Mississippi River Geomorphology and Potamology (MRG&P) Program, Project 470711. The MRG&P Program is part of the Mississippi River and Tributaries Program and is managed by the U.S. Army Corps of Engineers (USACE), Mississippi Valley Division (MVD). Data on

lakes and reservoirs were obtained from USACE Nashville, Omaha, Pittsburgh, Rock
Island, St Paul, Tulsa, Vicksburg, Sacramento, and Los Angeles Districts. The authors
would like to acknowledge the invaluable contributions from Matthew W. Farthing, Scott
D. Christensen, Elissa M. Yeates, Kayla A. Cotterman, and James W. Lewis all ERDC and
L. Austin Auld with the USACE Nashville District.

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
