# Peer review of "Comparison of Generalized Non-Data-Driven Lake and Reservoir"

_Hydrology and Earth System Sciences, 2019_

## Referee Comment (RC1) · Anonymous Referee #1 · 3 Aug 2019

"Comparison of generalized non-data-driven reservoir routing models for global-scale hydrologic modeling", HESS 2019

This study compares two methods for non-data-driven reservoir routing in large-scale hydrologic models, namely those of Doll et al. (2003) and Hanasaki et al. (2006). The methods are compared using 60 reservoirs in the United States over a 6 year period. The sensitivity of method coefficients are examined, as well as daily and monthly timesteps. Performance is measured by KGE, R^2, and RMSE and found to vary under certain geographic and hydroclimatic settings.

The overall motivation is clear - reservoirs have a major influence on flow, and generalized methods are needed because systems often follow specific rules that cannot be compiled at global scales. The research questions are interesting and framed very well, and the 60-reservoir dataset serves as a nice case study. However, the rationale for comparing these specific methods, and the general insights that can be drawn from the comparison, are less clear. I recommend major revisions based on the following points.

1. While I understand the advantage of methods that require minimal input data, there are many other methods that could be part of this comparison, many of which are mentioned in the literature review. Focusing only on these two makes the study seem thin, especially given that the methods are fairly similar - both are a linear relationship with either reservoir inflow or storage. It is hard to draw any general conclusions from the comparison about which approach works better, or why - or whether another approach not studied here might be more appropriate in certain contexts.

The discussion of results is overly system-specific and does not manage to provide much of this insight. Without including more state of the art methods the paper cannot be a definitive statement about their performance.

Related to this, the distinction with data-driven models is not obvious considering both methods studied here are empirical relationships. The setup of the study does not preclude statistical fitting techniques, and many reservoir routing models are based on some form of regression or machine learning.

2. If I understand correctly, these methods were originally intended for the monthly timescale. Testing at a daily timescale is an interesting experiment, but the results suggest only small improvements over the run-of-river baseline. Neither of these models can account for short-timescale operations such as flood control or hydropeaking. The reservoirs where the models perform well seem to have lower impoundment ratios.

Daily timestep routing is clearly a challenge, and not one that the authors would need to resolve in this paper. My concern is that the models are applied outside of their

intended use, which may have some consequences for how their performance is interpreted. For example, what are the implications for large-scale hydrologic modeling if these reservoir models do not outperform a run-of-river assumption?

3. How would these single-reservoir case studies apply to routing in a larger grid cell in a hydrologic model, where the runoff may incorporate outflows from several reservoirs as well as unregulated tributaries?

4. Is there a relationship between the operating objective of the reservoir (hydropower, flood control) and the routing model performance?

5. It is not clear why the study would compare the model performance prior to calibrating the k coefficients for each reservoir. I would think readers would be most interested in the performance results after calibration.

6. Finally, the paper could be edited to improve the flow of ideas.

---

## Referee Comment (RC2) · Anonymous Referee #2 · 6 Aug 2019

Gutenson et al. submitted a manuscript to HESS with the title "Comparison of Generalized Non-Data-Driven Reservoir Routing Models for Global-Scale Hydrologic Modeling" where they assess the performance of two reservoir operational schemes at daily and monthly resolution and for 60 reservoirs in the US with the aim to find a proper method for implementing in a large scale hydrological forecast model. While the manuscript gives interesting insights esp. into the sensitivity to the outflow coefficient of the method used by Döll et al. (2003) (hereafter referred to D03) and to the daily / monthly time step behaviour, I do have doubts with the relevance and novelty of this manuscript and have several comments as listed below.

[Figure]

Relevance and novelty:

The authors use two approaches that are around in the field of global hydrological modelling since more than a decade, and are in the meantime somehow outdated. A clear motivation (along with explanations and citations) is missing in the introduction that frames why it is required in the purpose of this manuscript (for hydrological forecast models) to use data-free schemes. Is it the specifics of hydrological forecast models (which is a way contradicting to the future outlook section where the authors indicate that assimilation schemes would be possible – if this is the case, why could not improved reservoir operation schemes be included)? In the publication of D03, "reservoirs are treated like global lakes, due to lack of information on their management" (Döll et al., 2003, p 112), hence it is not a reservoir algorithm per se. Having said that, it is true, that this approach is indeed data-free (except maximum storage). However, the Hanasaki et al. 2006 approach (hereafter referred to as H06) is not data-free (information about storage capacity, purpose, water demand downstream are required). Since Döll et al., 2009, the H06 approach was adapted and implemented into their GHM. Nowadays the most GHMs have further advanced reservoir schemes and consider e.g. also reservoir operation years (see www.isimip.org) and the reservoir schemes of some models have been evaluated by Masaki et al. (2017). Again, it is hard to understand, why the future of dealing with reservoirs in hydrological forecast models (as the authors indicated in Section 3.8) should lay in an algorithm that was not developed specifically for reservoirs. Current state of the art in reservoir operation schemes is much advanced since this two approaches (e.g. see the citing articles of H06 and Döll et al., 2009) and nowadays initiatives like http://globaldamwatch.org/ try to make the best out of available global scale information about reservoirs. Nevertheless, the research questions are well formulated and there are some very interesting technical aspects of the manuscript such as the sensitivity of to the outflow coefficient of D03 method and the time step assessment but it is questionable if this is worth it to publish in such a widely framed journal like HESS, mainly because those approaches (esp. D03) are outdated. I would encourage the authors to include (among the suggestions

below) more up-to-date approaches in a potential revision? That could widen up the usability of the findings of the work to e.g. the global hydrological modelling community.

Other major issues:

Essential to the performance measures of the approaches are inflow and outflow streamflow data from the 60 reservoirs. The inflow data were back-calculated from outflow data and storage changes (lines 509 ff). This is too vague and needs much more details, otherwise it is a black box and nothing that is reproducible. Furthermore, the authors have not quantified the uncertainty or plausibility of this back-calculation (only that inflows can be sometimes negative which is a sign that the back-calculation misses essential details) which must be definitely included. As a first guess, the authors should back-calculate the inflow of reservoirs from Nashville district with the same method as for the others and compare this to observed inflows (that are available when I am interpreting line 510 correctly).

From the title of the manuscript, it is not clear that the manuscript is motivated from the perspective of hydrological forecast models. Especially the first paragraphs of the introduction irritated me with respect to the title (global hydrological modelling). I therefore suggest to rephrase the title to better reflect the focus of the manuscript to specific needs of forecast models.

The sensitivity study of k_rd of the D03 method is very interesting and coming to the conclusion that the suggested parameter (0.01) is not optimal for most of the reservoirs, but 0.9 is. A factor of 0.9 means that 90% of the actual storage volume is being released by each time step. This seems to be – on a daily time step – very high, mimicking nearly a complete flow trough of inflow to outflow of the reservoirs. An analysis of k_rd in relation to IR would be very meaningful. I assume that those high k_rd values should occur only with low IR (or low S_t values) so that the reservoir only little modifies the river flow. If this is not the case, I would see that as indicator, that the D03 method simulates the river flow modification well for a false reason. In addition, Fig 5 mentions

"maximise KGE and minimize nRMSE" – which values are typical for e.g. the k_rd of 0.9? Close to 1 or rather close to or below 0? A maximum KGE of 0.095 (as displayed in Fig 6) is indeed maybe the maximum for this reservoir but I would not see this low efficiency metric as sign for good modelling result. I suggest therefore to use the classifications of the 4 KGE levels (lines 212 f) as stacked bar for better interpretation of Fig. 5. The major purpose of the 60 reservoirs is flood control (for 43 reservoirs). This goes well along the assumption that for H06 method only the non-irrigation purpose is being used (by the way, there are global (monthly) irrigation estimates available from global hydrological models, e.g. Huang et al., 2018)). Could the authors please assess more in detail how the methods analysed relates to flood events (in particular here, the uncertainty information of the back-calculation approach would be required)? This could test the two approaches if it holds true for such events.

The manuscript reads in principle well but the mix of considering all reservoirs and the focus of some for specific analysis is not very clear, probably because a clear difference between a results and discussion section is missing. What are the criteria to select specific reservoirs for focus analyses (e.g. selection of the 7 dams in Table 2)? For the discussion, I would suggest to read Masaki et al., (2017) with the aim of trying to relate your results to those of their study (they also dealt with e.g. Fort Peck), that could place your study better to recent literature.

Section 3.7: The authors state, that only H06 includes withdrawals in their method. While not completely wrong, this is in a way misleading. The analysed approach of D03 relates only to the outflow of the global lake / reservoir. However, in the D03 paper section 3.5 details of how water abstraction is considered from reservoirs / global lakes. So, water use is considered in the storage equation of their model and hence indirectly in the outflow calculation (as this is impacted by actual storage). The same holds true for evaporation (for both approaches). Lines 504-508 needs to be therefore rewritten to avoid misleading conclusions.

Section 3.8. seems to be contradicting. On the one hand, the authors argue that e.g.

the D03 method will be implemented in a river routing model, on the other hand, they argue that data-driven approaches (assimilation of remote sensing products) could be the future. What is the general message then? What about recent implementations of reservoir algorithms in the global hydrological models? Could they be implemented in river routing models? How does other routing models, e.g. CaMa-Flood deal with reservoirs?

Minor and formal issues (not complete):

At various places in the manuscript, the authors use "Döll Method, Hanasaki Method" in various different writing styles. I suggest to use abbreviations throughout instead (e.g. D03 / H06) for better readability and consistency.

Table 1 gives insights into the statistics of the reservoirs used for testing. However, it would be very informative to have those kind of statistics for every reservoir, including the coordinates and purpose, e.g. at appendix or as supplement. That could help interpreting the other figures e.g. Fig 7. I would also suggest to include the performance metrics for each reservoir and method (daily and monthly time step) to this table which increases interpretation possibilities (e.g. as excel file for downloading). Please also provide numbers of the reservoirs to Fig. 1 to relate the reservoir characteristics and interpretation to specific locations of the US.

The introduction contains many relatively old references (e.g. the effect of reservoir regulation to streamflow) that could be enriched with more recently published work.

Units or dimensions are missing in the equations

Unit "cms" in discharge time series figures should be written as $m^3$ s-1

Fig 6. is a "best" performance of KGE <0.1 meaningful at all?

Fig 7. Is very interesting – for which purpose is this dam created (see also my comment to Table 1)?

Fig 8: X-Axis labels not very common (sugget to use months only); why is the KGE (0.3 for Union City similar for wet and dry years and the simulation (same as 1.1 for Arcadia lake) whereas it is not the case for inflow/outflow assessment? The y-axis title is strange ("Outflow") as content shows also "Observed Inflow". I suggest for this (and similar diagrams) another title (e.g. streamflow or river discharge)

Fig 9b wrong header (should be daily?). Furthermore, could the authors explain, why in Fig 9a there is a low peak in monthly D03 time series within 2011 (is it the instability mentioned in Sect. 3.6?)? Is k_rd the same for Fig. 9a and 9b?

Fig 10 and the interpretation would highly benefit if the initial simulated outflow (without the adaption of Eq 5) is displayed as well.

541: Maybe I have missed it – but have the authors somewhere assessed the suitability at sub-daily time steps? If not, please modify this bullet point. Furthermore, the authors assessed only non-irrigation reservoir algorithm from H06. The whole study design is intended to handle non-irrigation purposes (except one reservoir) so it is unknown how the approach works in reservoirs that are constructed for irrigation purposes. This should be made clear as well.

References: some Discussion papers cited (Coerver et al 2017, Döll et al., 2009) which needs to be replaced by final published versions, formally the reference list is far away from being consistent, needs carefully revisions. I have not checked if all citations from the manuscript are listed in the references and vice versa.

References

Döll, P., Kaspar, F., and Lehner, B.: A global hydrological model for deriving water availability indicators: model tuning and validation. Journal of Hydrology, 270 (1-2), 105-134, https://doi.org/10.1016/S0022-1694(02)00283-4, 2003.

Döll, P., Fiedler, K., and Zhang, J.: Global-scale analysis of river flow alterations due to water withdrawals and reservoirs, Hydrol. Earth Syst. Sci., 13, 2413-2432,

https://doi.org/10.5194/hess-13-2413-2009, 2009.

Hanasaki, N., Kanae, S., and Oki, T.: A reservoir operation scheme for global river routing models, J. Hydr., 327 (1-2), 22-41, https://doi.org/10.1016/j.jhydrol.2005.11.011, 2006.

Huang, Z., Hejazi, M., Li, X., Tang, Q., Vernon, C., Leng, G., Liu, Y., Döll, P., Eisner, S., Gerten, D., Hanasaki, N., and Wada, Y.: Reconstruction of global gridded monthly sectoral water withdrawals for 1971–2010 and analysis of their spatiotemporal patterns, Hydrol. Earth Syst. Sci., 22, 2117-2133, https://doi.org/10.5194/hess-22-2117-2018, 2018.

Masaki, Y., Hanasaki, N., Biemans, H., Müller Schmied, H., Tang, Q., Wada, Y., Gosling, S. N., Takahashi, K., and Hijioka, Y.: Intercomparison of global river discharge simulations focusing on dam operation—multiple models analysis in two case-study river basins, Missouri–Mississippi and Green–Colorado. Environ. Res. Lett., 12, 5, 055002, https://doi.org/10.1088/1748-9326/aa57a8, 2017.
* * *

---

## Referee Comment (RC3) · Anonymous Referee #3 · 7 Aug 2019

The manuscript compares two generalized non-data-driven methods for reservoir routing in large-scale hydrologic models. Given the increase in large-scale hydrological models this is indeed a scientifically relevant question and of interest for developers as well as users of large-scale hydrological models. The paper evaluates two methods presented in previous papers using data from 60 reservoirs located in the US and provides overall model performances, sensitivity analysis, effect of time steps on model performance, model stability, and limitations. The paper is overall well structured, the research questions are clearly described and the language is precise. There are, however, in my view two major limitations that strongly limit the general insights of this manuscript and that require a major revision.

[Figure]

1.) Reservoir data used in this study:

This study uses 60 reservoirs in the US out of which 43 (71% of the used reservoirs in this study) are used for flood control and only 6 (10%) for hydropower generation and one (2%) for irrigation. Given that the majority of reservoirs worldwide are used either for hydropower production or water supply/irrigation the reservoirs selected in this manuscript are clearly NOT representative of the majority of reservoirs influencing river flow. This strongly limits the insights that this manuscript provides especially with regard to the research questions outlined in the introduction. Flood control reservoirs are arguably the reservoir type that is "easier" to simulate in a hydrological model as their operation is mostly correlated with reservoir storage levels. The electricity-demand-driven hydropower reservoirs or the water supply/irrigation reservoirs have a much stronger human intervention and depend on a variety of factors which are typically very difficult to capture in non-data-driven reservoir routing model. The authors have to ensure that there is at least an equal share of reservoirs for flood control, hydropower production and water supply/irrigation used in their analysis to be able to draw more general conclusions. An even better approach would be to make the specific analysis of model performance for different reservoir types.

2.) The authors have limited their study to the comparison of two non-data-driven methods. Given that both methods are already more than 10 years old and more and more data on reservoirs is becoming available (e.g. see http://globaldamwatch.org/ or the numerous publications to measure reservoir water level fluctuations from space) I would encourage the authors to include at least either one data driven method as a comparison or to include at least a reservoir model that allows for the possibility to include more data about the reservoir such as the Burek et al (2013) model. Otherwise this is again a strong limitation on the insights provided in this manuscript.

Minor comments:

1.) There are a number of spelling errors and english grammar errors that need to be

corrected.

2.) References Macia-Sorribes and Pulido-Velazquez 2017 is missing

3.) Page 3, lines 53-54: This is not correct. The Global Flood Awareness System GloFAS is accounting for reservoir influences in its forecasts.

4.) page 12, lines 255: Please explain better why a decrease in RMSE is observed in Fig. 2 using the Doell method while at the same time the method increases KGE and R-squared?

5.) Please explain why you did not calibrate the k coefficients for each reservoir in the Doell method. Or at least provide an analysis of the model performance calibrated versus uncalibrated k coefficient.

---

## Author Comment (AC1) · 27 Sep 2019

Thank you for your review and thoughtful discussion! Broadly, we will make alterations to the manuscript to more clearly describe the rationale for comparing these specific methods. We will also clarify some of the discussion and insights.

Issue #1 Response: The reviewer is correct in that there are a number of reservoir routing methods available that we did not choose to study. We describe why we chose these methods in lines 134-141 of the manuscript. We heavily reviewed existing reservoir routing methods and limited our study to Döll et al. (2003) and Hanasaki et al. (2006) because these are parsimonious approaches that require only readily available

[Figure]

Interactive
comment

variables, reservoir storage and inflow. Unlike, recent methods such as Burek et al. (2013) and Zajac et al. (2017), the Döll and Hanasaki models do not require a number of operational states. We describe these assumptions in lines 84-104. We will alter the manuscript to better describe how our study's objectives are linked to utilizing the most parsimonious approaches available and that the limits to this study are linked to our emphasis on being parsimonious. We will investigate additional methods, beyond those listed in the literature review, and either use these methods in the study or describe why they were not chosen as well. The reviewer is also correct concerning the pairing of non-data-driven methods with statistical fitting techniques. Data-driven methods are described in lines 66-78 of the manuscript. They are approaches that require a training dataset in order to be implemented. Non-data-driven methods are described in Lines 80-83. No doubt other approaches not studied here could be more appropriate in certain contexts, however, the primary aim here is evaluating methods for use in hydrologic forecasting schemes applicable across the global domain. The authors do not assume that subsets of training data are available to characterize operations, nor do they assume that real-time insights related to current reservoir levels can be known in a forecast setting. Non-data-driven methods are conceptualizations of reservoir operations that can be adapted to be a data driven approach but do not require training data in order to be implemented. We will alter the manuscript to describe that non-data-driven methods can be linked to statistical fitting techniques but that they are capable of being employed independent of such pairings.

Issue #2 Response: The run-of-the-river assumption is what we consider our "baseline" approach (lines 217-221). In other words, the utility of either method is based on whether the method outperforms the run-of-river assumption. And the timestep (daily vs. monthly) is does affect performance as discussed in Section 3.5. In general, the results might suggest negligible improvements over run-of-river for the Hanasaki et al. (2006) scenario at the daily timescale, as discussed in this paper. Alternatively, the improvement over the baseline condition, even at the daily timestep, was generally positive for the Döll et al. (2003) method when the release coefficient is adjusted for

the daily timestep. See Section 3.1 for a description of this. We note that there are limitations to the implementation of Döll et al. (2003) through the discussion of system specific examples. We also note in Section 3.6 the potential issue for instability. Thus, we discuss the model limitations. We kindly disagree that these models cannot in all cases adjust for flood control or hydropeaking. There are a small subset of simulations that perform worse than the baseline simulation using Döll et al. (2003). We will alter the manuscript to analyze and discuss the causes for this underperformance.

Issue #3 Response: This is beyond the scope of this manuscript. The use of observed inflow is a proxy for this scenario. A follow up study would be required to analyze this.

Issue #4 Response: We cannot make an assertion either way about the operating objective influencing the routing performance since the vast majority of reservoirs we considered are for flood control (see Section 2.2). The authors will note this limitation in Section 3.7.

Issue #5 Response: Calibration of the k coefficients would be better suited for the insertion of the Doll Method into a hydrologic routing scheme. The current study is investigating the feasibility of these methods and is a precursor to an additional study were the methods may be implemented in a large scale hydrologic routing scheme. In this study, we may calibrate the routing scheme using the k coefficients. However, there is limited benefit from calibrating the k coefficients in this study, given that it is an initial investigation into whether varying the k coefficients is beneficial. In addition, reservoir outflow information is rarely available at global scales, calibrating the k coefficients for an operational forecasting model would be very difficult (see the discussion in Zajac et al. (2017) of an open access database for daily reservoir records). When outflow information is available, the authors agree that it is advisable to calibrate. However, the authors do assume this type of information is available globally. We will include these details within the revised manuscript.

Issue #6 Response: We will alter the manuscript to improve the flow of ideas.

References:

Burek, P., Knijff, J. v. d., & Roo, A. de. (2013). LISFLOOD: Distributed Water Balance and Flood Simulation Model. Luxembourg, Belgium. https://doi.org/10.2788/24719

Döll, P., Kaspar, F., & Lehner, B. (2003). A global hydrological model for deriving water availability indicators: Model tuning and validation. Journal of Hydrology, 270(1–2), 105–134. https://doi.org/10.1016/S0022-1694(02)00283-4

Hanasaki, N., Kanae, S., & Oki, T. (2006). A reservoir operation scheme for global river routing models. Journal of Hydrology, 327(1–2), 22–41. https://doi.org/10.1016/j.jhydrol.2005.11.011

Zajac, Z., Revilla-Romero, B., Salamon, P., Burek, P., Hirpa, F. A., & Beck, H. (2017). The impact of lake and reservoir parameterization on global streamflow simulation. Journal of Hydrology, 548, 552–568. https://doi.org/10.1016/j.jhydrol.2017.03.022

---

## Author Comment (AC2) · 27 Sep 2019

Thank you for your review! Broadly, we will make alterations to the manuscript to clarify and broaden its applications.

Relevance and novelty: The authors use two approaches that are around in the field of global hydrological modelling since more than a decade, and are in the meantime somehow outdated. A clear motivation (along with explanations and citations) is missing in the introduction that frames why it is required in the purpose of this manuscript (for hydrological forecast models) to use data-free schemes. Is it the specifics of hydrological forecast models (which is a way contradicting to the future outlook section

where the authors indicate that assimilation schemes would be possible – if this is the case, why could not improved reservoir operation schemes be included)? In the publication of D03, "reservoirs are treated like global lakes, due to lack of information on their management" (Döll et al., 2003, p 112), hence it is not a reservoir algorithm per se. Having said that, it is true, that this approach is indeed data-free (except maximum storage). However, the Hanasaki et al. 2006 approach (hereafter referred to as H06) is not data-free (information about storage capacity, purpose, water demand downstream are required). Since Döll et al., 2009, the H06 approach was adapted and implemented into their GHM. Nowadays the most GHMs have further advanced reservoir schemes and consider e.g. also reservoir operation years (see www.isimip.org) and the reservoir schemes of some models have been evaluated by Masaki et al. (2017). Again, it is hard to understand, why the future of dealing with reservoirs in hydrological forecast models (as the authors indicated in Section 3.8) should lay in an algorithm that was not developed specifically for reservoirs. Current state of the art in reservoir operation schemes is much advanced since this two approaches (e.g. see the citing articles of H06 and Döll et al., 2009) and nowadays initiatives like http://globaldamwatch.org/ try to make the best out of available global scale information about reservoirs. Nevertheless, the research questions are well formulated and there are some very interesting technical aspects of the manuscript such as the sensitivity of to the outflow coefficient of D03 method and the time step assessment but it is questionable if this is worth it to publish in such a widely framed journal like HESS, mainly because those approaches (esp. D03) are outdated. I would encourage the authors to include (among the suggestions) more up-to-date approaches in a potential revision? That could widen up the usability of the findings of the work to e.g. the global hydrological modelling community.

Response: This reviewer is clearly approaching this paper from a climate modeling perspective. While there are commonalities between forecasting models and climate models, there are some distinctions that might not be appreciated. The climate model operates at half degree resolution on monthly timesteps and runs decadally. A flood forecasting model driven by a numerical weather model is more highly resolved spatially and temporally to capture individual flood events. The reviewer suggested initiatives like global dam watch which provides precisely the type of information needed to implement the non-data-driven D03 & H06 approaches, e.g. active storage volumes and total storage capacity. The majority of the more sophisticated approaches the reviewer alludes to require site specific operational rule curves or training data which are not contained in the Global Dam Watch's GRaDv1.3 database that has these attributes for 7,300 dams, which is still incomplete considering there are 38,660 dams geolocated in the GlObal geOreferenced Database of Dams (GOOD2). Likewise, the global (monthly) irrigation estimates by Huang et al. (2018) mentioned below are difficult to disaggregate at the spatial (100m -12km) and temporal scales (hourly to daily) of the forecasting models. Currently, a common practice in large-scale spatially continuous forecasting systems like the NOAA National Water Model is to neglect reservoirs altogether. Thus any approach that outperforms run-of-the-river conditions represents an improvement. The approaches the reviewer considers 'outdated' happen to be the most readily implementable, for instance the CaMa Model the review mentions was paired with H06 ( The authors will consider developments from Masaki et al. (2017) in their revised manuscript.

Other major issues: Essential to the performance measures of the approaches are inflow and outflow streamflow data from the 60 reservoirs. The inflow data were back-calculated from outflow data and storage changes (lines 509 ff). This is too vague and needs much more details, otherwise it is a black box and nothing that is reproducible. Furthermore, the authors have not quantified the uncertainty or plausibility of this back-calculation (only that inflows can be sometimes negative which is a sign that the back-calculation misses essential details) which must be definitely included. As a first guess, the authors should back-calculate the inflow of reservoirs from Nashville district with the same method as for the others and compare this to observed inflows (that are available when I am interpreting line 510 correctly).

Response: A true directly observed inflow is not available for nearly all reservoirs,

including those maintained by the U.S. Army Corps of Engineers. There are two ways that one can acquire an inflow; estimated using a streamflow model (as in Masaki et al., 2017) or use a back calculated inflow. The authors have chosen to utilize a back calculated inflow because this accounts for all other withdraws from the reservoir, such as irrigation, seepage, etc. This allows the authors to focus exclusively on the reservoir routing methodology utilized. This is also the reason that we do not account for such withdraws as this would be double counting withdraws. We will add details about back calculated inflow to the manuscript.

From the title of the manuscript, it is not clear that the manuscript is motivated from the perspective of hydrological forecast models. Especially the first paragraphs of the introduction irritated me with respect to the title (global hydrological modelling). I therefore suggest to rephrase the title to better reflect the focus of the manuscript to specific needs of forecast models.

Response: The authors will adapt the title and manuscript to better reflect the application of the paper to hydrologic forecast models.

The sensitivity study of k_rd of the D03 method is very interesting and coming to the conclusion that the suggested parameter (0.01) is not optimal for most of the reservoirs, but 0.9 is. A factor of 0.9 means that 90% of the actual storage volume is being released by each time step. This seems to be – on a daily time step – very high, mimicking nearly a complete flow trough of inflow to outflow of the reservoirs. An analysis of k_rd in relation to IR would be very meaningful. I assume that those high k_rd values should occur only with low IR (or low S_t values) so that the reservoir only little modifies the river flow. If this is not the case, I would see that as indicator, that the D03 method simulates the river flow modification well for a false reason. In addition, Fig 5 mentions "maximise KGE and minimize nRMSE" – which values are typical for e.g. the k_rd of 0.9? Close to 1 or rather close to or below 0? A maximum KGE of 0.095 (as displayed in Fig 6) is indeed maybe the maximum for this reservoir but I would not see this low efficiency metric as sign for good modelling result. I suggest therefore to use the

classifications of the 4 KGE levels (lines 212 f) as stacked bar for better interpretation of Fig. 5. The major purpose of the 60 reservoirs is flood control (for 43 reservoirs). This goes well along the assumption that for H06 method only the non-irrigation purpose is being used (by the way, there are global (monthly) irrigation estimates available from global hydrological models, e.g. Huang et al., 2018)). Could the authors please assess more in detail how the methods analysed relates to flood events (in particular here, the uncertainty information of the back-calculation approach would be required)? This could test the two approaches if it holds true for such events.

Response: It is correct that the DO3 calculation is a proportion of current storage over the time step. However, It should not be inferred that a $k_r$ of 0.9 leads to a 90% release of total storage volume. A second factor using the proportion of active storage taken to the 3/2 power decreases the proportion of storage released for a given time step (Equation 4 on line 196). The intent of D03 is to simulate outflow approaching inflow when available storage is low and inflows are high. This is the type of outflow decision that will be made during such cases as, in reality, spillways will begin to open and outflow will approach, or exceed, inflow. We will undertake and add an analysis of $k\_rd$ in relation to IR to the revised manuscript. Our results do not suggest the KGE of 0.095 is a good modeling result. However, we do substantially improve model performance, over run-of-the-river, for the majority of the cases we analyzed. The 0.095 KGE referenced by the reviewer is an example from the Yazoo Basin Headwaters Project used to transparently illustrate a case in which the non-data driven models do not perform well. This site specific analysis represents an attempt to analyze why DO3 performs poorly under certain instances. In fact, we consider the reason why the performance is low for this particular reservoir in Section 3.3. A stacked bar graph is a great idea for inclusion into the manuscript. The authors will add this to the manuscript.

The manuscript reads in principle well but the mix of considering all reservoirs and the focus of some for specific analysis is not very clear, probably because a clear difference between a results and discussion section is missing. What are the criteria to

select specific reservoirs for focus analyses (e.g. selection of the 7 dams in Table 2)? For the discussion, I would suggest to read Masaki et al., (2017) with the aim of trying to relate your results to those of their study (they also dealt with e.g. Fort Peck), that could place your study better to recent literature.

Response: The dams included in Table 2 were selected to illustrate that conclusions for reported by Hanasaki et al. (2006) at the monthly timestep do not necessarily hold when the impoundment ratio is such that outflow bears minimal relation with inflow. This will be clarified in the revised manuscript. The authors will analyze Masaki et al. (2017) and attempt to relate the results of this study with that of Masaki et al. (2017). The authors are willing to segregate Section 3 (Results and Discussion) into a Results Section (currently Sections 3.1 & 3.2) and a Discussion Section (currently 3.3-3.7) if the editors feel this provide additional clarity.

Section 3.7: The authors state, that only H06 includes withdrawals in their method. While not completely wrong, this is in a way misleading. The analysed approach of D03 relates only to the outflow of the global lake / reservoir. However, in the D03 paper section 3.5 details of how water abstraction is considered from reservoirs / global lakes. So, water use is considered in the storage equation of their model and hence indirectly in the outflow calculation (as this is impacted by actual storage). The same holds true for evaporation (for both approaches). Lines 504-508 needs to be therefore rewritten to avoid misleading conclusions.

Response: The reviewer is correct that the D03 study does implicitly account for water withdrawals in the hydrologic simulations which they perform. This accounting of water withdrawals is similarly accounted for in the inflow that our study uses. The inflow that our study uses is back calculated from storage fluctuations. However, the D03 equation has no explicit term for withdrawals, a portion of the H06 formulation does. We will clarify these differences within our manuscript.

Section 3.8. seems to be contradicting. On the one hand, the authors argue that e.g.

the D03 method will be implemented in a river routing model, on the other hand, they argue that data-driven approaches (assimilation of remote sensing products) could be the future. What is the general message then? What about recent implementations of reservoir algorithms in the global hydrological models? Could they be implemented in river routing models? How does other routing models, e.g. CaMa-Flood deal with reservoirs?

Response: Our intention in Section 3.8 was to illustrate how remotely sensed data could be assimilated into non-data-driven methods to improve them. Unlike data-driven methods, the non-data-driven methods we consider are conceptualizations of reservoir operations that can be adapted to be a data driven approach, but do not themselves require training data in order to be implemented. We see this as an advantage. The authors will clarify this point in the manuscript by explaining that non-data-driven methods can be linked to statistical fitting techniques, but that they are capable of being employed independent of such pairings. CaMa-Flood has been coupled off-line with H08 (Mateo, 2014), an integrated water resources model that includes the H06 reservoir operation module evaluated herein (Hanasaki et al., 2008).

Minor and formal issues (not complete): At various places in the manuscript, the authors use "Döll Method, Hanasaki Method" in various different writing styles. I suggest to use abbreviations throughout instead (e.g. D03 / H06) for better readability and consistency.

Response: The authors will update the manuscript to refer to the Döll Method and Hanasaki Method as D03 and D06, respectively.

Table 1 gives insights into the statistics of the reservoirs used for testing. However, it would be very informative to have those kind of statistics for every reservoir, including the coordinates and purpose, e.g. at appendix or as supplement. That could help interpreting the other figures e.g. Fig 7. I would also suggest to include the performance metrics for each reservoir and method (daily and monthly time step) to this table which

[Figure]

increases interpretation possibilities (e.g. as excel file for downloading). Please also provide numbers of the reservoirs to Fig. 1 to relate the reservoir characteristics and interpretation to specific locations of the US.

Response: The authors will update the manuscript to reflect these requested changes.

The introduction contains many relatively old references (e.g. the effect of reservoir regulation to streamflow) that could be enriched with more recently published work.

Response: The authors will update the manuscript literature review to include works such as Masaki et al. (2017).

Units or dimensions are missing in the equations

Response: The authors will update the manuscript equations to include units and dimensions.

Unit "cms" in discharge time series figures should be written as m3 s-1

Response: The authors will update the manuscript to reference discharge as m3 s-1

Fig 6. is a "best" performance of KGE.

Response: The authors will add update to Fig. 6. to describe the figure as the "best" performance of KGE.

References:

Döll, P., Kaspar, F., and Lehner, B.: A global hydrological model for deriving water availability indicators: model tuning and validation. Journal of Hydrology, 270 (1-2), 105-134, https://doi.org/10.1016/S0022-1694(02)00283-4, 2003.

Döll, P., Fiedler, K., and Zhang, J.: Global-scale analysis of river flow alterations due to water withdrawals and reservoirs, Hydrol. Earth Syst. Sci., 13, 2413-2432, https://doi.org/10.5194/hess-13-2413-2009, 2009.

Hanasaki, N., Kanae, S., and Oki, T.: A reservoir operation scheme for global river routing models, J. Hydr., 327 (1-2), 22-41, https://doi.org/10.1016/j.jhydrol.2005.11.011, 2006.

Huang, Z., Hejazi, M., Li, X., Tang, Q., Vernon, C., Leng, G., Liu, Y., Döll, P., Eisner, S., Gerten, D., Hanasaki, N., and Wada, Y.: Reconstruction of global gridded monthly sectoral water withdrawals for 1971–2010 and analysis of their spatiotemporal patterns, Hydrol. Earth Syst. Sci., 22, 2117-2133, https://doi.org/10.5194/hess-22-2117-2018, 2018.

Mateo, C., N. Hanasaki, D. Komori, K. Yoshimura, M. Kiguchi, A. Champathong, T. Sukhapunnaphan, D. Yamazaki, and T. Oki (2013), A simulation study on modifying reservoir operation rules: Tradeoffs between flood mitigation and water supply, in Considering Hydrological Change in Reservoir Planning and Management, edited by A. Schumann (IAHS Publ. 362) pp. 33– 40, IAHS Press, Wallingford, U. K.

Mateo, C.M., N. Hanasaki, D. Komori, K. Tanaka, M. Kiguchi, A. Champathong, T. Sukhapunnaphan, D. Yamazaki, T. Oki. Assessing the impacts of reservoir operation to floodplain inundation by combining hydrological, reservoir management, and hydrodynamic models. Water Resources Research, vol.50, pp.7245-7266, 2014, doi:10.1002/2013WR014845

Masaki, Y., Hanasaki, N., Biemans, H., Müller Schmied, H., Tang, Q., Wada, Y., Gosling, S. N., Takahashi, K., and Hijioka, Y.: Intercomparison of global river discharge simulations focusing on dam operation at multiple models analysis in two case-study river basins, Missouri–Mississippi and Green–Colorado. Environ. Res. Lett., 12, 5, 055002, https://doi.org/10.1088/1748-9326/aa57a8, 2017.

---

## Author Comment (AC3) · 27 Sep 2019

Thank you for your thoughtful commentary! We will utilize your comments to clarify the applicability of this study and clarify our discussion of why we utilize the chosen reservoir routing schemes.

Response to General Comment #1: Though the majority of reservoirs in our study are primarily flood control, most are multipurpose and used for both flood control, irrigation, recreation, and hydropower. For example, the reservoirs along the Missouri River are also hydropower facilities. Interestingly, the Yazoo Basin Headwaters Projects are arguably more focused on flood control and yet the methods we use in this study are

incapable of capturing the release behavior of these reservoirs. In order to broaden our evaluation to include a more diverse collection of dams, the authors would need historical inflow and discharge from for a comparable time period. We are not aware of any sizable databases that contain this information. Should this data exist to support this, the authors would gladly include this in their analysis. The authors will make it clear in the manuscript that the reservoirs in this study are almost exclusively multipurpose and perform more than flood control.

Response to General Comment #2: There are numerous studies that measure reservoir level fluctuations from space, many of these studies focus on measuring the areal extents of reservoirs, not predicting reservoir outflow. See for example Nguy-Robertson et al. (2018). The authors are aware of initiatives like Global Dam Watch referenced by the reviewer (e.g.http://globaldamwatch.org/) which provides precisely the type of information needed to implement the non-data-driven Doll & Hanasaki approaches, e.g. active storage volumes and total storage capacity. The majority of the more sophisticated approaches require site specific operational rule curves or training data which are not contained in the Global Dam Watch's GRaDv1.3 database which contains these attributes for 7,300 dams, a small fraction of the 38,660 dams geolocated in the GlObal geOreferenced Database of Dams (GOOD2).

We did not include the Burek et al. (2013) and Zajac et al. (2017) methods because of the strong assumptions that are made concerning storage capacity limits and naturalized streamflow thresholds (see lines 84-104) which are less parsimonious limiting their utility for forecasting. No doubt other approaches not included here could be more appropriate in certain contexts, however, the primary aim here is evaluating methods for use in hydrologic forecasting schemes applicable across the global domain. The authors do not assume that subsets of training data, i.e. historical discharge, are available to characterize operations, nor do they assume that real-time insights related to current reservoir levels can be known in a forecast setting. Non-data-driven methods are conceptualizations of reservoir operations that can be adapted to be a data driven

approach but do not require training data in order to be implemented. We will alter the manuscript to describe that non-data-driven methods can be linked to statistical fitting techniques but that they are capable of being employed independent of such pairings.

Response to Minor Comment #1: We will go through the manuscript for any spelling or grammatical errors.

Response to Minor Comment #2: We will add these references to the manuscript.

Response to Minor Comment #3: This is true, Zajac et al. (2017) discuss this. GLoFAS is limited to 463 of the largest lakes and 667 largest reservoirs out of 33,000 large dams (>15 m high) registered with ICOLD. Limitations include 1) some information necessary for parameterization and validation of lake and reservoir routines is not available in the GRanD database suggested by two of the reviewers; 2) reservoir records for deriving case-specific operation rules (and related model parameters) are not readily shared. The two approaches (Doll & Hanasaki) are meant to address these limitations which contribute to considerable uncertainty around parameter values described by Zajac et al. (2018) that adversely affects model performance. We will correct this in the manuscript.

Response to Minor Comment #4: A decrease in Root Mean Squared Error is natural when model accuracy improves (i.e., KGE and R-Squared). The authors will add this information to the manuscript.

Response to Minor Comment #5: Calibration of the k coefficients would be better suited for the insertion of the Doll Method into a hydrologic routing scheme. The current study is investigating the feasibility of these methods and is a precursor to implementation in a large-scale hydrologic routing schemes. There is limited benefit to calibrating the k coefficients in this study, given that reservoir outflow information is rarely available at global scales to calibrate against so that calibrating the k coefficients operationally would be challenging without release records (see the discussion in Zajac et al. (2017) of an open access database for daily reservoir records). We will include these details

within the manuscript.

References:

Burek, P., Knijff, J. v. d., & Roo, A. de. (2013). LISFLOOD: Distributed Water Balance and Flood Simulation Model. Luxembourg, Belgium. https://doi.org/10.2788/24719

Nguy-Robertson, A., May, J., Dartevelle, S., Birkett, C., Lucero, E., Russo, T., . . . Zentner, M. (2018). Remote Sensing Applications : Society and Environment Inferring elevation variation of lakes and reservoirs from areal extents : Calibrating with altimeter and in situ data. Remote Sensing Applications: Society and Environment, 9 (December 2017), 116–125. https://doi.org/10.1016/j.rsase.2018.01.001

Zajac, Z., Revilla-Romero, B., Salamon, P., Burek, P., Hirpa, F. A., & Beck, H. (2017). The impact of lake and reservoir parameterization on global streamflow simulation. Journal of Hydrology, 548, 552–568. https://doi.org/10.1016/j.jhydrol.2017.03.022
* * *

---

## Author Response (AR1)

Dear Editor and Reviewers:

My coauthors and I thank you for you thoughtful insight into how we can improve our manuscript, which is now entitled, "Comparison of Generalized Non-Data-Driven Lake and Reservoir Routing Models for Global-Scale Medium-Range Hydrologic Forecasting of Reservoir Outflow at Diurnal Time Steps". Generally, our edits have improved the flow of ideas and clarified the discussion and insights gained from this analysis.

Specific improvements that have been made include:
1. The authors updated the manuscript to refer to the Döll Method and Hanasaki Method as D03 and H06, respectively.
2. The authors adapt the title and manuscript to better reflect the application of the paper to hydrologic forecast models at daily time steps.
3. The authors evaluated Masaki et al. (2017) to determine if their results at reservoirs along the Missouri River were comparable to those in this study. Because the study is more focused on intermodel comparison at seasonal time steps, there is little overlap with the intentions of our study and no comparison of the manuscripts deemed necessary by the authors.
4. We have verified that all inflow estimates in our reservoir sample are a back calculated inflow.
5. To better describe why a back calculated inflow was used in our study, Section 2.1 now describes why a back calculated inflow was chosen in this study. Section 3.7 describes the limitations of this study, based upon the use of a back calculated inflow.
6. To better describe our study's objectives, clarification of why the D03 and H06 methods where chosen was provided in Section 1.2.
7. The manuscript was altered in Section 3.8 to better describe that non-data-driven methods can be linked to statistical fitting techniques and remote sensing data.
8. We investigate the reservoir routing methodology employed by Wada et al. (2014) but do not include this method because we deem it to be too simple and too similar to the Döll et al. (2003) approach. Section 1.2 describes this investigation in the manuscript.
9. In Section 1.2, we alter the manuscript to more clearly describe the rationale for comparing D03 and H06.
10. Units and dimensions were added to the descriptions of the equations in Section 2.1
11. Added the reference Macian-Sorribes and Pulido-Velazquez (2017) to the listed references.
12. A statement was added to Section 3.1 to explain why RMSE decreases and R-Squared and KGE increase.
13. Figure 6, Figure 7, Figure 8, Figure 9, and Figure 10 were altered to reference discharge as $m^3 s^{-1}$.
14. We have reviewed the document for spelling and grammatical errors.
15. A stacked proportional bar graph (Figure 4) and analysis were added to Section 3.1 to better describe the improvement that D03 provides over the baseline and H06 simulations.
16. We added verbiage to Section 2.2 make it clear in the manuscript that the reservoirs in this study are almost exclusively multipurpose and perform more than flood control.

17. An analysis of best performing k_rd in relation to IR was conducted and no significant statistical or visual relationship was found.
18. The authors found only one instance where model accuracy was substantially worse than the baseline condition. We consider this to be an outlier in our study because this reservoir behaves much differently than reservoirs of a similar IR and average inflow. We note this in Section 3.1 of the manuscript.
19. In Section 3.7, we added a discussion concerning the lack of diversity in reservoir operational purposes in our study's sample and how this inhibits the study's ability to determine the effect purpose has on reservoir routing performance.
20. Clarification was added to Figure 2, Figure 3, and Figure 5 to ensure that the description captured that these simulations depicted describe the best performing form of D03 and H06.

We look forward to your feedback on this version of the manuscript. Thank you again for your time and patience.

Best,
Joseph Gutenson

[revised manuscript text omitted]

---

## Author Response (AR2)

Dear Dr. Wanders:

Below are Reviewer #4's comments. In bold, you should find our response to those comments.

Please let my coauthors and I know if you need any additional clarification.

Thank you,
Joseph Gutenson

Gutenson et al., performs a comparison of two parsimonious non-data driven reservoirs operation models. The statistical comparison across the models and baseline is appropriate and in general the paper is well written and clearly structured. Compared to the previous version of the manuscript I believe that the authors have addressed most of the concerns raised by the reviewers, although I was not among them. Yet, I have some remaining concerns that I consider should be addressed before accepting the paper for publication.

1) I understand that the authors have qualitatively addressed the choice of the two models D03 and H06 in the introduction due to their parsimony, but I feel that the comparison remains weak as it is. I think the conclusions of the study would be even stronger if the authors included also less parsimonious models, e.g., the Burek et al., 2013 or Zhao et al., 2016 (if data allows to apply them at the daily time step). This has the dual added value to inform the GHM community on potential tradeoffs between model complexity and performance, and shows in which cases a more complex reservoir operation model might be more appropriate.

**Response: The authors assert that a comparison between D03, H06, and more complex models is best suited for an additional manuscript. We agree that potential tradeoffs should be better understood. The function of the current article is to determine if a simple, parsimonious approach has utility in large-scale diurnal hydrologic forecasting at reservoirs. We have provided a qualitative description of why D03 and H06 have been chosen. We have improved our qualitative decision by creating Table 1 which describes various non-data-driven models and their associated inputs. In future work, the authors will determine how these parsimonious approaches compare with more complex models. We have noted these limitations in the Limitations section of the manuscript and have listed this future comparison in the Future Work section.**

2) I think that a breakdown of the results (especially Figure 2 and 4) based on dam purpose would be very valuable. If most of the dams are multipurpose and include hydropower, then plotting the residuals against the dam's installed capacity might give insights on the potential dependence of the model performance on hydropower potential operation (rather relevant at the daily time step).

**Response:  While nearly all dams in this study are multipurpose, 45/60 are primarily used for flood control, leaving statistically insignificant sample sizes of the remaining dams with differing primary purposes.  We do not analyze the effect of the dam's primary purpose because of the limitations of our current sample of dams.  An investigation of the effect of dam purpose on reservoir routing performance should be pursued with a more diverse sample of dams.  The authors have noted this within the Future Work section.  However, the current limitations of this sample preclude such analysis from providing fruitful results, analysis, and discussion in this work.**

3) The limitations section should address the sample bias and related implications. Might very well be that the D03 outperforms the H06 in temperate regions but everywhere else the H06 might be a better choice. For similar reasons I recommend to not implement solely the D03 in RAPID but also the H06 model to extend the performance testing in regions outside the US in future studies.

**Response:  A discussion of sample bias and related implications has been added to the Limitations section of the manuscript.**

Minor remarks:
L. 163-164 abbreviations D03 and H06 are introduced but are already present in the text prior to this point. Should be corrected.

**Response: This issue was resolved in the manuscript.**

[revised manuscript text omitted]